# Microbial Exopolysaccharide Composites in Biomedicine and Healthcare: Trends and Advances

**DOI:** 10.3390/polym15071801

**Published:** 2023-04-06

**Authors:** Vishal Ahuja, Arvind Kumar Bhatt, J. Rajesh Banu, Vinod Kumar, Gopalakrishnan Kumar, Yung-Hun Yang, Shashi Kant Bhatia

**Affiliations:** 1University Institute of Biotechnology, Chandigarh University, Mohali 140413, Punjab, India; 2University Centre for Research & Development, Chandigarh University, Mohali 140413, Punjab, India; 3Department of Biotechnology, Himachal Pradesh University, Shimla 171005, Himachal Pradesh, India; 4Department of Life Sciences, Central University of Tamil Nadu, Thiruvarur 610005, Tamil Nadu, India; 5Centre for Climate and Environmental Protection, School of Water, Energy and Environment, Cranfield University, Cranfield MK43 0AL, UK; 6Institute of Chemistry, Bioscience and Environmental Engineering, Faculty of Science and Technology, University of Stavanger, P.O. Box 8600 Forus, 4036 Stavanger, Norway; 7Department of Biological Engineering, College of Engineering, Konkuk University, Seoul 05029, Republic of Korea; 8Institute for Ubiquitous Information Technology and Applications, Seoul 05029, Republic of Korea

**Keywords:** biopolymers, exopolysaccharides, EPS composites, healthcare, food

## Abstract

Microbial exopolysaccharides (EPSs), e.g., xanthan, dextran, gellan, curdlan, etc., have significant applications in several industries (pharma, food, textiles, petroleum, etc.) due to their biocompatibility, nontoxicity, and functional characteristics. However, biodegradability, poor cell adhesion, mineralization, and lower enzyme activity are some other factors that might hinder commercial applications in healthcare practices. Some EPSs lack biological activities that make them prone to degradation in ex vivo, as well as in vivo environments. The blending of EPSs with other natural and synthetic polymers can improve the structural, functional, and physiological characteristics, and make the composites suitable for a diverse range of applications. In comparison to EPS, composites have more mechanical strength, porosity, and stress-bearing capacity, along with a higher cell adhesion rate, and mineralization that is required for tissue engineering. Composites have a better possibility for biomedical and healthcare applications and are used for 2D and 3D scaffold fabrication, drug carrying and delivery, wound healing, tissue regeneration, and engineering. However, the commercialization of these products still needs in-depth research, considering commercial aspects such as stability within ex vivo and in vivo environments, the presence of biological fluids and enzymes, degradation profile, and interaction within living systems. The opportunities and potential applications are diverse, but more elaborative research is needed to address the challenges. In the current article, efforts have been made to summarize the recent advancements in applications of exopolysaccharide composites with natural and synthetic components, with special consideration of pharma and healthcare applications.

## 1. Introduction

Exopolysaccharides (EPSs) are natural biopolymers synthesized by microorganisms and are secreted for various respective functions, such as defense, biofilm formation, pathogenicity, structure, adhesion, etc. [1]. EPSs are long-chain biomolecules with molecular weights ranging from 10 to 30 kD, and are produced during the late exponential and or stationary phase of microbial growth. EPSs are produced in response to environmental stress conditions, such as pH and temperature, and exposure to heavy metals or inhibitors, etc. [2,3]. Biochemically, EPSs are carbohydrate polymers composed of glucose, galactose, and rhamnose, accompanied by non-carbohydrate moieties such as proteins, enzymes, nucleic acid, etc. The exact composition may vary with microorganisms and growth conditions. 

Based on the biochemical structure and composition, EPSs can be categorized into homo-exopolysaccharides (HOEPSs) and hetero-exopolysaccharides (HEEPSs) [4]. HOEPSs are composed of one type of monosaccharide, such as α-D-glucans, β-D-glucans, fructans, and polygalactans, interlinked with α-1-6, α-1-3, β-1-2, β-1-3, β-2-6, and β-2-1 linkage among the subunits, depending upon the monomeric units. Dextran is one of the well-known examples of HOEPSs, which is made of glucose interlinked with α-1-6 glucoside linkage. In contrast, HEEPSs are comprised of different sugar monomers, along with their respective derivatives and non-carbohydrate moieties. Succinoglycan is one of HEEPSs present in bacterial biofilm. It is comprised of saccharide-based oligomers in which sugar molecules are derivatized with acyl, pyruvate, and succinic acid [4,5,6]. Both classes are further recognized as subgroups based on the dominant sugar residues [4]. In recent years, EPSs have gained wide attention for their cost-effective production and diverse applications as pharma and healthcare products, cosmeceutical, nutraceutical, functional food, and biocontrol agents in agriculture [7], oil recovery in petroleum industries [8], heavy metal removal [9], drug delivery, and tissue regeneration and repair [10]. Microbial exopolysaccharides have applications in various sectors, as depicted in Figure 1. 

Most exopolysaccharides have gained attention in healthcare due to their nontoxicity, and biocompatibility, but dextran is the only known exopolysaccharide that has been commercialized in healthcare as a plasma volume compensator [6,11]. Alongside that, the majority of applications in healthcare are still under trial. Prasher et al. [12] used a dextran derivative, i.e., acetylated dextran, as a drug delivery vehicle for the treatment of respiratory disease due to biodegradability, pH sensitivity, high encapsulation efficacy, and the ability to cross the mucosal layer. Yahoum et al. [13] encapsulated metformin hydrochloride in xanthan gum microspheres and found a sustainable release of metformin hydrochloride from the microsphere. The eyes are one of the most sensitive parts of the body that need special care. EPSs have proven safe and biocompatible for biological systems, which allows their application in ophthalmic formulations. Khare et al. [14] evaluated an ophthalmic solution comprising gellan-gum-based nanosuspension with posaconazole for fungal keratitis. The main bottleneck of using native EPS molecules in commercial products is solubility in different media, bioavailability, degradation, etc. Xanthan, a HEEPS comprising a glucose backbone along with trisaccharide side chains, has poor thermal stability and electrical conductivity and is prone to microbial contamination [15,16]. Curdlan has high immunomodulatory potential, good gelling ability, and thermal stability, but is disadvantaged by the issue of solubility in water [17]. Hyaluronic acid has high water retention, but poor mechanical stability [18,19]. EPS has been blended with other natural biomolecules or synthetic polymers to improve and control the functional features of base EPS molecules, such as solubility, antimicrobial potential, mechanical strength, water retention, etc. The approach has proven beneficial, as a hyaluronic acid (HA) composite has higher mechanical strength and stability. HA composites with poly(Ne-acryloyl L-lysine) have shown a double network structure and is used for the fabrication of a more physiologically relevant 3D in vitro model for breast cancer [20,21]. Similarly, a HA composite with a self-assembling peptide carrying an IKVAV adhesive motif was used for the fabrication of a scaffold for breast cancer [22], and composites with silk fibroin–gelatin and heparan sulfate was used for the scaffold fabrication for cholangiocarcinoma [23]. A chitosan–curdlan composite is characterized by important features of both EPS components, as chitosan forms a fibrillary scaffold, and curdlan provides support to mesenchymal cell adhesion and promotes bone growth [24].

EPS composites have shown a way to overcome the bottlenecks of native EPS molecules by offering 3D-stable and porous architecture, offering more support for cell adhesion and proliferation, and an overexpression of enzymes governing wound healing and mineralization. However, there are a lot of factors impeding the commercialization of composite materials. Among the major challenges, stability and activity after prolonged storage must be addressed for commercialization apart from the cost of the product. The current review summarizes the major applications of EPS composites in healthcare, as well as research challenges. 

## 2. Microbes Producing Exopolysaccharides

Exopolysaccharides are produced by diverse microorganisms, including yeasts, fungi, and bacteria, utilizing various raw materials (Figure 2). Besides the native physiological role, microbial EPSs also have diverse applications in many industries. Several efforts have been made by various researchers for the production of EPSs (Table 1). In comparison to yeasts and other fungi, probiotic bacteria are commonly used due to non-pathogenicity and categorization of generally regarded as safe (GRAS) [25]. It has been found that carbon source has a direct relation with EPS production and composition. A study on 20 strains of *Lactobacillus paracasei* revealed that with a change in carbon source, not only does EPS yield, but it also influences the monosaccharide composition. The yield of EPS was increased by 115% with optimized carbon sources, including fructose, glucose, galactose, lactose, mannose, and trehalose [26]. Among different carbon sources, including sucrose, maltose, lactose, glycerol, and sorbitol, maximum EPS production has been achieved with maltose by *Candida guilliermondii* and *Candida famata* (0.505 and 0.321, respectively) [27]. Among 156 lactic acid bacteria isolated from healthy young children’s feces, the maximum EPS production, i.e., 59.99 g/L, was reported from *Weissella confusa* VP30 after 48 h in growth media containing 10% sucrose [28].

The use of commercial-grade sugars/substrates has a direct impact on product cost, and is one of the factors responsible for high production costs. Hence low-cost waste materials, such as lignocellulosic residues and wastewater, are preferred as raw materials for EPS production. The carbohydrate and organic fractions present in the waste can be used by microorganisms. It opens up the opportunity to reduce product costs, along with waste management. Da Silva et al. [29] have compared coconut shells, cocoa husks, and sucrose for xanthan gum production by *Xanthomonas campestris* pv. *campestris* IBSBF 1866 and 1867. The study revealed that xanthan gum yields were higher, i.e., 4.48 g/L and 3.89 g/L, in the case of cocoa husk by *Xanthomonas* strains 1866 and 1867, respectively, but the apparent viscosity was higher than sucrose, i.e., 181.88 mPas over cocoa husk, with a viscosity of 112.06 mPas.

Choi et al. [30] used spent media wastewater (originated from kimchi fermentation) for EPS production using *Leuconostoc mesenteroides* WiKim32. Under optimal conditions, the maximum EPS productivity was 7.7–9.0 g/L, with a conversion of 38.6–45.1%. The EPSs were nontoxic and exhibited thermal tolerance and antioxidant activity. Pan et al. [31] optimized dextran production from *Leuconostoc pseudomesenteroides* XG5 using an L9 (33) orthogonal test. Under optimal conditions (sucrose 100 g/L, pH 7.0, 25 °C, at 100 rpm for 36 h), the maximum dextran yield of 26.02 g/L and 40.07 g/L were recorded at a laboratory scale and fed-batch fermentation, respectively. The EPS was also exhibiting water-holding capacity and antioxidant activity. It reduced the chewiness and hardness of yogurt, but the resilience increased during the 14 days of storage. Product cost is one of the main obstacles in commercialization, hence process economics is one of the major factors that must be assessed. Integration of multiple processes might improve process economics, as well as environmental adaptability. *Sphingobium yanoikuyae* was evaluated for the coproduction of EPS and polyhydroxyalkanoates (PHAs) using lignocellulosic hydrolysate. Hydroxymethyl furfural (HMF), one of the hydrolysis byproducts, improved the consumption of glucose and xylose during fermentation. The optimum C/N ratio of 5 resulted in the maximum EPS production of 3.24 ± 0.05 g/L, however, a further increase in the C/N ratio (30) favored PHB accumulation (38.7 ± 0.08% *w*/*w*) [32]. Biomass hydrolysate is usually accompanied by phenolics, furfurals, and HMF, which also act as fermentation inhibitors and hinder microbial action. Some of the methods, including activated carbon-based adsorption and membrane filtration, have been proposed for hydrolysate detoxification [33,34]. Removal of inhibitors might improve microbial action and product yield. Bhatia et al. [32] have compared the potential of various raw and detoxified hydrolysates for EPS production. In comparison to raw hydrolysate, detoxified biomass hydrolysate showed increased EPS production and maximum EPS production was reported with detoxified pine biomass hydrolysate, i.e., 2.83 ± 0.03 g/L. Table 1 summarizes the recent efforts for EPS production using various microbes (yeasts, bacteria, and fungi), utilizing various pure carbon and organic waste materials.

**Table 1 polymers-15-01801-t001:** Microbial production of EPS from various substrates.

EPS	Organism	Substrate	Growth Conditions	Working Volume	EPS Yield	Key Achievements	Reference
Dextran (α-D-gluco pyranosyl moieties interlinked with α-(1,6) linkage and have α-(1,2)/α-(1,3)/α-(1,4) branching) 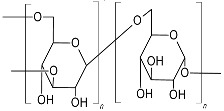	*Leuconostoc mesenteroides* SF3	10% sucrose	Temp 25 °C; pH 6; incubation period 16 h; inoculum 24 h old, 10% inoculum with cell concentration of 10^8^ cells/mL.	100 mL	23.8 ± 4 g/L	Water absorption capacity 361.8% ± 3.1; oil absorption capacity 212.0% ± 6.7; emulsion activity 58.3% ± 0.7.	[35]
	*Lactobacillus* spp.	15% sucrose	Temp 30 °C; pH 7; incubation period 24 h; inoculum 24 h old 4%, growth conditions aerobic.	100 mL	5.8 mg/mL	*Lactobacillus* strains were isolated from the human vagina and infant stool.	[36]
	*Leuconostoc pseudomesenteroides* DSM20193	Brewers’ spent grain	Initial cell concentration 6.0 Log cfu/g; temp 25 °C; period 24 h.	1000 g	1.2 g/100 g	EPS production is accompanied by mannitol; no dextran production without a starter (commercial granulated sugar).	[37]
	*Leuconostoc pseudomesenteroides* XG5	100 g/L sucrose	Temp 25 °C; pH 7.0; mixing rate 20 rpm; time period 60 h; inoculum 2%.	35 L	26.02 g/L dextran	Protein content in EPS reduced when extracted with EDTA or NaOH+formal-dehyde.	[31]
	*Weissella confusa* A16	Brewers’ spent grain	Initial cell concentration 6.0 log cfu/g; temp 25 °C; period 24 h.	1000 g	1.1 g/100 g	No mannitol production was observed, but a starter was required for EPS production.	[37]
Curdlan (Type HOEPS, unbranched; molecular weight 5.3 × 10^4^–2 × 10^6^ Da; components glucosyl residues inter-connected with β-D-(1→3) bonds) 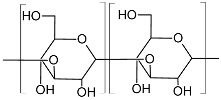	*Agrobacterium* sp. IFO 13140	50 g/L	Synthetic medium; temp 30 °C; mixing rate 150 rpm; period 5 d; pH 7.	100 mL	-	Water holding capacity and oil holding capacity 64% and 98% higher in comparison to commercial curdlan.	[38]
	*Bacillus cereus* PR3	10% starch	Synthetic medium; period 96 h.	100 mL	20.88 g/L	Anti-oxidant activity increased with curdlan.	[39]
	*Agrobacterium* sp. ATCC 31749	Asparagus spear bottom part juice	Synthetic medium; temp 30 °C; mixing rate 200 rpm; period 168 h.	100 mL	40.2 g/L	Curdlan production is higher with sucrose in comparison to mineral salt.	[40]
Xanthan (Type HEEPS; components backbone made of D-glucose unit linked with β-1,4-glycosidic bonds and side chain trisaccharide; side chain comprised of mannose, glucuronic acid, and mannose, terminal mannose with pyruvic acid residues; molecular weight 2.0 × 10^6^–2.0 × 10^7^ Da) 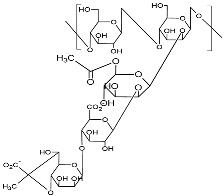	*Xanthomonas campestris*	20 g/L glucose	Stainless steel supported biofilm reactor; period 27 h; synthetic medium; mixing rate 180 rpm.	150 mL	3.47 ± 0.71 g/L	Use of biofilm reactor increased the xanthan recovery.	[41]
	*Xanthomonas campestris*	20 g/L glucose	Polyethylene supported biofilm reactor; period 78.5 h; synthetic medium; mixing rate 180 rpm.	150 mL	3.21 ± 0.68 g/L	Biofilm reactor increased the glucose consumption.	[41]
Gellan gum (Type HEEPS; components backbone made up of β-d-glucose, l-rhamnose, and d-glucuronic acid along with acetate and glycerate attached to glucose) 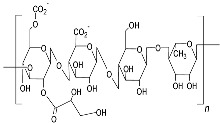	*Sphingomonas pseudosanguinis* (Accession No. GI:724472387)	80 g/L biodiesel-derived waste glycerol	pH 7; synthetic medium; temp 30 °C; mixing rate 200 rpm; inoculum 10%; period 7 days; 0.5 vvm.	3 L	51.6 g/L	At lower concentrations, glycerol is consumed completely at all pHs, but at a higher concentration, it is not exhausted completely.	[42]
	*Sphingomonas yabuuchiae* (GI:724472388)				52.6 g/L		[42]
EPS Br42 was found to be a heteropolysaccharide consisting of glucose and galacturonic acid with a molecular weight of about 286 kDa.	*Brevibacillus borstelensis* M42	2% glucose	Period 60 h.	-	1.88 g/L	Water-holding capacity 510 ± 0.35%, oil-holding 374 ± 0.12% and swelling capacities 146.6 ± 5.75%.	[43]
EPS K1T-9 (EPS type HEEPS; components glucose and galacturonic acid; molecular weight 207 kDa.	*Neorhizobium urealyticum* sp. nov.	Glucose 5 g/L	Zobell’s marine broth; pH 7; temp 28 °C; mixing rate 150 rpm; inoculum size 5 mL/100 mL; period 72 h.	Working volume 400 mL	3.38 g/L	Water holding 356 ± 0.8%, oil holding 697 ± 1% (coconut oil); 317 ± 1.3% (olive oil), swelling capacity 200 ± 1.1%.	[44]

Preparation of composites for healthcare applications needs high purity, therefore the downstream processing becomes an inseparable part of processing after fermentation. After production recovery, the identification of structural and chemical characteristics is necessary for further applications. The characterization of EPSs is quantitative, as well as qualitative. EPSs are mainly comprised of carbohydrates conjugated with other biomolecules, hence the basic characterization techniques employed are a colorimetric estimation and use spectrophotometry [45]. For carbohydrate estimation ‘Dinitrosalicylic Acid Reagent’ is one of the common methods which quantify the reducing sugars [46]. Similarly, for protein, Bradford’s dye-binding method [47] and Lowry’s method [48] are used. Besides basic characterization with colorimetric methods, Fourier transform infrared spectroscopy (FTIR) is employed to detect the available functional groups and structural functionalities of EPSs. For the detailed structures of EPSs, nuclear magnetic resonance (NMR) and mass spectra are used. In NMR, the sample is dissolved in deuterated solvents for quantification with respect to internal standards [49,50]. The mass spectrum of EPS provides a monosaccharide composition. For analysis, EPSs are hydrolyzed with acid hydrolysis, followed by silylation derivatization. The derivatives are detected and identified by gas chromatography-mass spectrometry [49]. The biological potential of EPS has followed the general procedure for antimicrobial, antioxidant, anti-inflammatory and other activities [45].

## 3. Microbial Exopolysaccharide Composites and Their Applications

Exopolysaccharides are natural biopolymers exhibiting diverse applications, variation in structure, adaptability, and the presence of different functional groups. These polymers have shown their suitability as drug carriers and medical sealants. Even with polymeric nature and biocompatibility, these polymers have also shown some weak points. Rapid degradation, hydrophilic nature, low mechanical and tensile strength, and stress tolerance, particularly for scaffold preparation, restrict their application. Some EPSs lacks bioactivity itself, which suggests the addition of multiple drug compounds to add bioactivity for in vivo application. Exopolysaccharide composites have shown a higher potential than native EPSs due to the presence of secondary polymer molecules that offer hybrid characteristics of both (Table 2). The higher mechanical strength with different functional characteristics contributes to diverse functional groups, and is responsible for physiological and chemical properties, contributed to by the members of the composite. To date, EPS composites have been prepared with natural, as well as synthetic, polymers. The selection of secondary polymers relies upon the type of application such as drug loading and release, tissue support and engineering, or ex vivo applications such as a sealant (Figure 3). 

Piola et al. [51] prepared a composite hydrogel with gelatin and xanthan gum to support the growth of human skin cells. The composite was printed with CellInk Inkredible 3D printer, using glutaraldehyde solution as a crosslinker. The printed hydrogel was compatible and suitable for the growth of human keratinocytes, as well as fibroblast. Alvel et al. [52] also prepared a composite hydrogel of xanthan with Konjac glucomannan, which also focused on wound healing. On the other hand, a 3D scaffold for tissue engineering was prepared with alginate–gellan gum [53] and methacrylated gellan gum [54]. 

The applications of EPS composites with various polymers have been summarized below:

### 3.1. Exopolysaccharide Composites with Natural Materials

EPS itself is a natural biopolymer that is combined with other natural biomolecules, including proteins, enzymes, lipids, carbohydrates, etc., and thus in most cases, the composite has healthcare-associated applications, including medical sealants and scaffolds in tissue recovery and repair. The major advantages of EPS composites with natural polymers are bioactivity, biodegradability, and biocompatibility. As both the components are natural in origin, there will be a negligible possibility for inflammation and rejection. Alongside that, these composites will not have any toxic effect on the host, but may also act as an antimicrobial formulation for targeting microorganisms. Most biological materials can also be produced in bulk amounts using microorganisms, hence the cost of the products is also reduced along with the higher availability. The addition of other polymers may also add a specific type of bioactivity, such as antimicrobial, anti-inflammatory, and even catalytic activity to the composite, which widens the possible application in different fields [55,56,57].

#### 3.1.1. Cellulose Composites with Natural Polymers

Cellulose is a linear structural polysaccharide comprised of glucan chains connected with cellobiose residues via β-1,4-glycosidic linkage. These structures are packed in forms of microfibrils, kept together with hydrogen bonds and Van der Waals interactions. Depending upon the packaging, it exhibits different degrees of polymerization. It is usually present in the secondary cell wall of plants, but is also present in some bacteria, including members of the Acetobacter, Agrobacterium, Azotobacter, Alcaligenes, Pseudomonas, Rhizobium, and Sarcina genera [58]. The main function of cellulose is to provide strength to structure [59], however, cellulose does not exhibit antimicrobial or similar bioactivity in itself [60], but its high mechanical strength and stress tolerance nature make it suitable for the fabrication of scaffold and dressing material when used together with some bioactive material. Various polymers have been exploited to form composites with EPSs, which improve their strength and water retention capacity and also aid in antimicrobial properties.

Ojagh et al. [61] produced cellulose with *Gluconacetobacter xylinus* by static fermentation. The cellulose and diethylaminoethyl cellulose were derivatized to carboxymethyl cellulose and carboxymethylated diethylaminoethyl cellulose, respectively, and processed for composite preparation. Briefly, cellulose is a carbohydrate that is rich in hydroxyl groups, which makes it negatively charged. However, similar charges on both components make interactions repulsive. Derivatization of cellulose to carboxymethylated diethylaminoethyl cellulose (CMDEAEC) has both hydroxyls, as well as amine groups in its structure. In contrast to cellulose, CMDEAEC has a net positive charge, and thus the higher amount of CMDEAEC can be loaded on BC in composite formation. The composite of bacterial cellulose–carboxymethyl cellulose has a higher drug loading capacity and swelling ratio than the composite of bacterial cellulose–carboxymethylated diethylaminoethyl cellulose, and even native bacterial cellulose itself. The drug release follows the Higuchi and Korsmeyer–Peppas models. The model is most suitable to describe the release of drugs from the polymer matrix hydrogel. The model suggests that drug release increases from the carrier with time. Methylene blue is a positively charged molecule and the presence of cellulose supports its binding to the composite. The work has also shown that chemical modification adds unique properties to EPSs, improves functionality, and enhances the interaction with another polymer to form a stable composite. Bacterial cellulose was functionalized by derivatization with two active agents, i.e., glycidyl trimethylammonium chloride and glycidyl hexadecyl ether. These agents act upon hydroxyl functional groups of glucose by a heterogeneous reaction, which simultaneously deprotonates the hydroxyl group, as well as the addition of epoxides. It was observed that mere derivatization reduced the bacterial population of *Staphylococcus aureus* 6538PTM and *Escherichia coli* (Migula) ATCC^®^ 8739TM almost by half (53% and 43%, respectively) within 24 h upon direct contact. However, the derivative did not have any cytotoxic effect in terms of morphology and viability upon keratinocytes (HaCaT cell line) and almost 90–100% viability was recorded after 6 days of direct contact. Modified hydrogel has shown an equitant wound closure rate in an in vitro scratch assay, with complete coverage of the wound area after 5 days [60]. It was suggested that the addition of epoxide to cellulose adds antimicrobial properties to cellulose that are also exhibited by the composite.

Composite hydrogels are made by attaching TEMPO-modified nanocrystalline cellulose to the methacrylated gelatin backbone. The human adipose-derived mesenchymal stem cells were encapsulated within the composite and cultured in normal and osteogenic media for 14 days, and the expression of valve interstitial cell phenotypes was observed. The encapsulated cells have lowered alpha-smooth muscle actin expression, while the expression of vimentin and aggrecan increased. Cells cultured in osteogenic media have a reduced expression of osteogenic genes (Runx2 and osteocalcin) that support resistance to calcification. With the composite, a tall and self-standing tubular structure was constructed with a composite hydrogel that also sustained cell viability for possible application in cardiovascular systems [62]. The addition of cellulose to chitosan has improved the mechanical strength, porosity, cytocompatibility, and drug release rate for use in bone tissue engineering. The ternary complex of alginate, chitosan, and bacterial cellulose was used for the preparation of scaffolds using a hydroxyapatite/D-glucono-δ-lactone complex-based gelling system. The composite-based scaffold can adapt to 3D morphology and is stabilized with extensive cross-linking. The smaller size of pores supports the attachment and growth of tissue, along with the required mechanical integrity. Alginate controls the swelling behaviors of the scaffold by intermolecular hydrogen bonds and prevents the degradation of the composite. It has shown high protein adsorption and release potential, along with commendable cytocompatibility, that supported its application in tissue engineering [63]. The polymer supported the 3D structure and allowed for the proliferation of cells, increasing the possible application opportunities in the reconstruction of complex and large organ structures. 

#### 3.1.2. Dextran Composites with Natural Polymers

Dextran is a non-toxic homoexopolysaccharide, mostly produced by lactic acid bacteria. It is made up of glucose subunits interconnected with α-(1→6) bonds, and forms a linear backbone. However, liner chains also have D-glucose side chains as branches attached with α-(1→4), α-(1→3), or α-(1→2) linkage. Dextran exhibited diversity in molecular weights that range from 40–70 kDa. Dextran is water soluble, increases viscosity, and has a tendency to expand in the presence of water, contributing to the volume expansion in the case of plasma volume compensation, gelling agent in foods and pharmaceuticals [6,11]. It is also used in the preparation of molecular sieves while purifying macromolecules [20]. The solubility and nontoxicity of dextran have a big role in its application for drug delivery and preparation of wound dressing materials for biological systems. 

Dextran was oxidized with periodate-assisted oxidation, followed by a reaction with chitosan hydrochloride. The composite did not have any cytotoxicity and offered minimal swelling in phosphate-buffered saline, along with good adhesiveness. The optimum adhesive strength of the composite, i.e., 200–400 gf/cm^2^, was 4–5-fold higher than commercial fibrin glue. The burst strength of the composite was 400–410 mm of Hg, which makes it suitable to be used as a medical sealant to control bleeding during surgical procedures. It also works under high blood pressure conditions. Both, adhesiveness and hemostat were assessed and found to be suitable in a rabbit liver injury model [64]. Besides medical adhesiveness, dextran has shown commendable suitability for in vivo applications, such as drug delivery and wound healing. Dextran composites alone can contribute to both adhesiveness and drug-carrying capacity. Even though a number of drugs are available for wound dressing and tissue repair, in some cases, injury worsens due to failure of wound dressing, mainly attributed to poor bioavailability, hydrophobic nature of drugs, and high level of reactive oxygen species. Dextran-based composites have offered the solution for all these challenges of rapid tissue repair. The drug delivery and bioavailability can be improved further when used in the nano form due to increased surface area. Andrabi et al. [65] have fabricated a composite nano-hydrogel with gelatin, oxidized dextran, and curcumin and cerium oxide-based nano-formulation. Two types of dextran derivatives have been used, i.e., alkylated dextran with 1-bromohexadecane (O-hexadecyl-dextran) and oxidized dextran. Curcumin nanoparticles were synthesized with alkylated dextran, while, on the other hand, cerium oxide nanoparticles were loaded onto the gelatin hydrogel with oxidized dextran. Altogether, these formulations form a hybrid nano-hydrogel that is used for wound healing. Here, alkylated dextran was amphiphilic in nature, which maintains the bioavailability of the drug (curcumin). Cerium oxide acts as an antioxidant and controls the reactive oxygen species and other free radicals. Hydrogel has shown a prolonged and consistent release of the drug, with ~63% release in 108 h. It also promotes cell migration to induce wound healing, along with antioxidant and in vivo anti-inflammatory activity (~39%).

Poor wound healing raises concerns in cases of noncompressible injuries due to a higher rate of blood loss. These injuries mostly occur from gunshots or injuries with sharp objects, such as knives, etc. In the absence of timely repair of the injury, higher blood loss and trauma-assisted mortality have been observed. Cryogels were prepared from oxidized dextran, chitosan, collagen, and polydopamine nanoparticles. In the formulation, dextran was used as a crosslinker, while chitosan was a hemostatic agent. The cryogels have an intensively branched and interconnected macroporous structure, with a high capacity to absorb blood or water. In vitro assessment suggested a very high coagulation potential due to strong procoagulant ability, and high adhesion potential for fibrinogens and blood cells. Moreover, it can activate platelets and associate intrinsic pathways. It reduced bleeding time and blood loss [66]. The dextran–thyme magnesium-doped hydroxyapatite composite is prepared by mixing salt precipitate with dextran and essential oil solution (Figure 4). The composites have shown antimicrobial activity and have considerable results against *Staphylococcus aureus*, *Enterococcus faecalis*, *E. coli*, *Pseudomonas aeruginosa*, and *Candida albicans.* The composites have the potential for application as antimicrobial coating materials [67].

#### 3.1.3. Xanthan Composites with Natural Polymers

Xanthan is a hetero-exopolysaccharide, comprising glucose, mannose, and glucuronic acid. It is produced and secreted by *Xanthomonas campestris*. Biochemically, it is made of repeated units of pentasaccharides interconnected with β 1, 4-linked d-glucose backbone. The chain also has a substitution with trisaccharide side-chain linkage. The monosaccharide composition represents β-D-glucose, α-D-mannose, and α-D-glucuronic acid in a 2:2:1 ratio. Xanthan is known for its high viscosity, emulsion stabilization, and shear-thinning activity. Xanthan is non-toxic and biocompatible, but it is disadvantaged by poor electrical conductivity and stability of heat [20]. In comparison to macrostructures, the application of nanomaterials increases the interactions at a molecular level, has a higher surface area, and reduces the drug dosage required due to the higher efficiency of drug delivery in the nano form [68,69]. The application of EPS composites as nanomaterials might improve the performance, hence efforts have been made to incorporate the nanomaterials into the composite. 

Nanocomposites comprising sodium alginate and xanthan gum, reinforced with cellulose nanocrystals and/or halloysites, are highly porous, with an extensive pore network. Nanotubes and nanocrystals have uniform dispersion and partial orientation within a composite. Composites with nanocrystals and nanotubes have porosity ranging from 91.7 ± 0.81% to 88.5 ± 0.64%, and water uptake capacity ranging from 14.73.7 ± 0.46 g/g to 11.34 ± 0.32 g/g. The composite is thermally stable and has high compressive strengths of 91.1 ± 1.2 kPa to 114.4 ± 0.6 kPa in dry form, and 9.0 ± 0.8 kPa to 10.6 ± 0.8 kPa in wet form. The composite has high cytocompatibility for MC3T3-E1 osteoblastic cells, and the viability of the cells increased with nanotube components in the composite. Both nitrocellulose and nanotubes increased the mechanical stability of the composite, along with conducive bioactivity, including higher cell adhesion and proliferation. The compressive strength of the composites was higher than alginate alone, as well as the alginate-xanthan gum blend, i.e., 91 kPa and 80.7 kPa. The blending of xanthan gum lowered the stiffness, which was further improved by the addition of nanocrystals and nanotubes, with the maximum strength reaching 114.4 kPa [70]. Mechanical strength and adhesion are important parameters for wound dressing material. Hydrogels of xanthan gum and Konjac glucomannan blend have shown significant firmness and cohesiveness when both polymers are used in an equal ratio, i.e., 1:1. All the blends prepared with components of 1:1 and 2:3 ratios (glucomannan: xanthan gum) have a water content of more than 99% and water contact angle of less than 90°. This feature is also important, as it prevents dehydration in wounds and supports healing by improving cell adhesion. In vitro analysis suggested that even after 72 h of contact, the hydrogel did not affect the morphology and viability, and supported the proliferation of fibroblasts. The hydrogel also provides cell migration during the healing phase that aided in rapid wound healing, which led to the full coverage of the wound site after 12 h, while in the control, it took 24 h [52].

Medical adhesives have been used in device and tape manufacturing. Biocompatible adhesive might offer a durable and user-friendly alternative for application. Feng et al. [71] have developed medical-grade adhesives with soybean protein sources at different concentrations of xanthan gum. The composite comprising 0.5% xanthan and 0.5% soybean proteins have shown a maximal adhesion strength of 321 kPa, which was 2.6-fold higher than the control, i.e., SP alone. The addition of xanthan increased the viscosity of the composite and improved the hydrogen bond. On a molecular basis, the addition of xanthan reduced the α-helix content and increased the β-sheet content in the protein secondary structure. The composite has shown a compact and viscous surface that supports adhesion. Sometimes, bone injuries need support and artificial implants for regeneration and recovery. Zia et al. [72] have prepared a polymer composite with xanthan gum, chitosan, and nanohydroxyapatite and polyelectrolyte complex. Osteo-conductive tri-composite scaffolds were prepared for osteo-regeneration. The composite scaffold, containing polymer electrolyte and hydroxyapatite in the ratio of 1:1, exhibited a commendable porous structure with a compressive strength swelling ability with slower degradation rates. Analysis in simulated body fluid, xanthan gum, chitosan, and nano-hydroxy apatite have apatite-like surface structures. In vitro interaction studies revealed that the nanocomposite scaffold with chitosan and nanohydroxyapatite supported cellular viability, attachment, and proliferation of MG-63 cells.

Polymeric porous scaffolds were made from chitosan and xanthan, along with 5% hydroxyapatite and brushite, for advanced mesenchymal stem cells. Composite scaffolds have an amorphous structure phase, having major bands of amide for chitosan and xanthan. Scaffolds have porous structures with calcium phosphate fillers. The elasticity modulus was higher for composite scaffolds with brushite than with hydroxyapatite and composite alone. Composites alone have a higher cell viability than scaffold–calcium phosphate, having acceptable cell viability. The composite with any calcium phosphate forms had a higher inflammatory response after 48 h, while scaffold + hydroxyapatite with mesenchymal stem cells had the lowest inflammatory cell number. It was clear from the work that calcium phosphate improved mechanical strength, but lowered cell viability. The toxic effect was countered by the addition of mesenchymal stem cells [73]. Now, the era of 3D printing has started, but the fabrication of biological structures needs supportive and efficient materials with a high mechanical strength and elasticity modulus. Piola et al. [51] developed crosslinked 3D-printable hydrogel with gelatin and xanthan gum, especially for wound dressing. The results have suggested that both gelatin and xanthan gum are important for composite formation, but xanthan concentration is crucial for the printability of the composite, as 1–1.2% of xanthan is required to attain printability, irrespective of the concentration of gelatin; however, 1.2% concentration was optimum.

#### 3.1.4. Pullulan Composites with Natural Polymers

Pullulan, an unbranched homopolysaccharide, is composed of triose units made from three glucose units interconnected by α-1,4 glycosidic bonds. These maltotriose units are linked with each other by α-1,6 glycosidic bonds. It is produced by *Aureobasidium pullulans* from starch, and secreted out [74,75]. The α-bond contributes to its aqueous solubility and flexibility. Overall, it is a biodegradable, nontoxic, and biocompatible polymer [76]. The characteristics are suitable for hydrogels and hybrid polymer fabrication, but the main drawbacks are its high-water solubility and hygroscopic nature. In addition, it has poor support for cell adhesion and proliferation, and is poorly osteogenic, mainly attributed to its hydrophilicity [77,78].

For making bone grafts, mechanical strength and degradation profile are important parameters. Usually, a hydrogel has a porous structure with intense networking. Pullulan is one of the non-immunogenic biopolymers that can be used for scaffold fabrication. Pullulan–dextran composite scaffolds, together with interfacial polyelectrolyte complexation fibers, have an improved adhesion for cells in comparison to pullulan. Further addition of extracellular proteins proved beneficial for cell adhesion and growth. The composite scaffold induced endothelial cell growth and followed the zero-order release kinetics for bovine serum albumin, as well as vascular endothelial growth factor [79]. Hydrogels are made from pullulan and covered with 5% hydroxyapatite nano-crystals and 3% poly(3-hydroxybutyrate). Making composites with fillers, such as butyrate, increased the compressive modulus of the composite scaffold by 10 times. The adhesion support for cells was improved by the presence of hydroxylapatite nanocrystals [78].

Posterior segment eye diseases need invasive intravitreal-injection-assisted treatment. However, prolonged injections become painful and can be countered with the use of an efficient drug delivery system. Kicková et al. [80] prepared pullulan–dexamethasone conjugates for sustainable drug delivery. Dexamethasone was loaded onto pullulan in a 1:20 ratio, and attached with pullulan via hydrogen bonds. The composite particles were stable at a wider range of temperatures, from 4 to 37 °C, at a physiological pH. At pH 5.0, these composites were acting like lysosomes and released the drug slowly, as 50% of the drug was released in the vitreous, while at pH 5.0, this occurred across a 2-day period. In vitro evaluation of biocompatibility showed no signs of toxicity on the retinal pigment epithelial cell line (ARPE-19).

Sustainable drug release from the composites has also opened the path for designing matrices and scaffolds for wound healing. The research work of Chen et al. [81] has proved that dressing materials have a critical role in wound healing time, pattern, and cellular response on wound healing, from in-depth histologic and histopathologic analysis in mice models. The dressing materials made from collagen hydrogel followed a human-like wound-healing model in mice. In comparison to the control, the hydrogel of pullulan–collagen induced rapid wound closure and healing, healing with less dense and shorter collagen fibers with random alignment.

Baron et al. [82] also used a pullulan composite for wound healing and drug delivery. Hydrogels made from oxidized pullulan and dopamine have possible applications in hemostatic wound dressing. Cryogels were prepared by (hemi)acetal and Schiff base bonds between dialdehyde pullulan and dopamine. Two types of formulation were prepared, i.e., PD1 by pullulan derivatives attached with dopamine derivatives and PD2 by adsorbed dopamine on the scaffold. PD1 is loaded with 20% dopamine, while PD2 has only 1.14% carried within the polysaccharide network. The porosity was maximum in the scaffold alone (80.41%) and reduced after loading; however, the network density was maximal in PD1 (47.1%). The water adsorption capacity was minimum for the crude scaffold (31.41%) due to the smooth texture and the maximum PD1 (59.01%). The PD1 hydrogel exhibited fast swelling initially, followed by stabilization. In comparison to the crude scaffold and PD2, PD1 has the highest mechanical stability of 558 N. The in vitro hemolytic analysis suggested a high rate of hemolysis in PD2, i.e., 99.04%, followed by 7.12% in PD1 and a minimum for the crude scaffold (0.15%).

#### 3.1.5. Levan Composites with Natural Polymers

Levan and gellan are exopolysaccharides extensively used in the food industry. Levan is a bioactive, user-friendly, non-toxic homo-exopolysaccharide produced by microorganisms, as well as plants. Among microorganisms, *Aerobacter levanicum*, *Bacillus subtilis*, *B. polymyxa*, *Corynebacterium laevaniformans*, *Pseudomonas* sp., and *Streptococcus* sp. are among some common producers of levan. It is one of the natural fructans, made of fructose subunits interlinked by β-(2→6) glycosidic linkages, along with side chains attachment via β-(2→1) linkage to the main backbone. It is commonly used as a thickening/gelling agent, encapsulating material, and alternative to petrochemicals in medical applications [83]. Levan is soluble in water, as well as in oil. In industries, it is used as gum, sweetener, flavor carrier, surface finishing additive, gelling agent, emulsifier, cryoprotector, and osmoregulator. Its bioactivity includes antitumor and antihyperlipidemic and radioprotector activities [84]. 

Levan has an immunomodulatory effect on the host, hence it is commonly used as a drug carrier, as well as drug coating material. Bovine serum albumin was used as a model compound, and was encapsulated in levan nanoparticles to assess carrying and release behavior. The nanoparticle has surface charges ranging from +4.3 mV to +7.6 mV, varying in particle size, from 200 nm to 537 nm. With the increase in size, the encapsulation capacity of nanoparticles also increased from 49.3% to 71.3%. Both types of particles have shown controlled release of BSA during in vitro analysis [85]. Besides drug delivery, levan has also been used as tissue filler. Due to its flexibility, an injectable biofiller was designed with levan, together with carboxymethyl cellulose and Pluronic F127, for the regeneration of soft tissue. The hydrogel offered an elastic modulus higher than hyaluronic acid hydrogel. The presence of interconnected pores also makes it suitable to be used as a filler. In vitro analysis has confirmed that levan improved cell proliferation and collagen synthesis in human dermal fibroblast cells without any cytotoxicity. The levan hydrogel was stable over 2 weeks in vivo, which was higher than the Pluronic F127 hydrogel or hyaluronic acid hydrogel alone. Apart from this, the levan hydrogel has also shown anti-wrinkle activity in wrinkle model mice, which was also higher in comparison to the hyaluronic acid hydrogel [86]. The hydrolysed derivative of the levan polysaccharide is prepared for nanoparticle synthesis by the electro-hydrodynamic atomization (EHDA) technique, and loaded with resveratrol (encapsulation efficiency 13.8 ± 1.3%). The drug-loaded nanoparticles have followed first-order kinetics in resveratrol release at different pHs, as higher loading is accompanied by a more gradual release. It showed a burst release mechanism, as 65–70% of the drug was released initially, followed by slow release. There was no sign of toxicity during in vitro assessment with human dermal fibroblast cell lines (PCS-201-012) [87]. Activation of metalloproteinase supports the healing of injured tissue. The application of levan in biomedical dressing sealants is limited, as it gets removed in a wet environment. The addition of catechol improved the adhesion strength of the levan composite to 42.17 ± 0.24 kPa (>3 times of fibrin glue), that persists even in a wet environment, and allowed for its application in hemostatic surgeries and wound healing. Besides adhesion, it also induced rapid blood clotting and healing of rat skin incisions. In comparison to levan, the composite has a lower endotoxin level [88]. The major challenge with levan is its cost and lower yield, thus the cost of composites and prepared products will also be high. For better market reach, bulk and cost-effective production is necessary.

#### 3.1.6. Gellan Composites with Natural Polymers

On the other hand, gellan gum is a hetero exopolysaccharide composed of two β-D-glucose, β-D glucuronic acid, and α L-rhamnose subunits. It is an anionic, water-soluble polysaccharide, secreted by *Sphingomonas elodea*. Gellan gum is a strong gelling agent, able to form a gel at very even low concentrations. Based on the gel-forming ability, two types of gellan gums are available, i.e., low-acyl form that makes hard and brittle gel, and the second high-acyl form, that produces soft and elastic gels. It can be used to make low-viscosity suspensions, and as a gelling agent in industries [74]. It can offer diverse forms in gelling ability, but it lacks stability against stress and shear tolerance for tissue engineering. Zheng et al. [89] have blended gelatin with gellan gum to prepare injectable scaffolds applicable for skin regeneration. The aim was to cover irregularly shaped wounds followed by recovery. Gelatin and gellan gum had a synergetic effect on the composite hydrogel and exhibited both shear-thinning, as well as self-recovering abilities. Further addition of tannic acid induced rapid wound healing in mice model.

Some of the studies have also shown the application of biopolymer composites as a cell carrier in addition to drug compounds, as suggested in the case of retinal pigment epithelium cells. The deterioration and damage in retinal pigment epithelium cells lead to blindness that can be cured with the replacement of damaged cells with healthy ones. Kim et al. [90] have shown that hydrogels prepared from gellan gum and silk sericin can be used as a cell carrier also. Sericin improved the compressive strength of composite over gellan gum to support the growth and proliferation of cells. A composite of gellan gum with 0.5% sericin has shown compressive strength to 10 kPa and improved the gene expression associated with ARPE-19 cell proliferation.

### 3.2. Exopolysaccharide Composites with Synthetic Polymers

Similar to natural polymers and biomolecules, synthetic polymers have also been used to make a composite with exopolysaccharides, which improves the binding ability and woven strength of any fiber. The section summarized some of the composites of exopolysaccharides and synthetic polymers. 

#### 3.2.1. Dextran Composites with Synthetic Polymers

Dextran composites are prepared with synthetic polymers mainly to improve the water/solvent interaction behaviors, biodegradability, and mechanical strength. In the context of medical/healthcare-associated operations, polymers have higher strength or compounds with additional health benefits have a critical contribution. The fabrication of advanced fabric with biodegradability and inbuilt healing power might improve the treatment and revolutionize healthcare practices. A biocompatible dressing material fiber was prepared with electrospun poly(vinyl alcohol)–dextran, prepared from dextran and poly(vinyl alcohol) by using citric acid as a cross-linker, followed by loading with sodium ampicillin. The mechanical strength was mainly governed by the concentration of citric acid, as it determines the degree of cross-linking. However, swelling, adsorption of protein, and drug release were decreased, as the CA concentration increased, while high concentrations of dextran induced the proliferation of HFB-4 cells and offered higher antimicrobial activity. After 24–48 h of treatment, all the fabrics had the potential to accelerate the wound gap closure [91]. Kenawy et al. [91] used a composite of poly(vinyl alcohol)–dextran nanofibers for injury dressing materials. The composite was prepared by poly(vinyl alcohol) and dextran cross-linked with sodium ampicillin-loaded citric acid. The composite was electrospun for the fabrication of dressing material. The composite made with 10% PVA–10% dextran and 5% citric acid offered the best nanofiber suitable for dressing materials. The concentration of the cross-linker, i.e., citric acid, greatly influenced the characteristics, including mechanical stability, thermal stability, and water uptake. The composite nanofibers with a high concentration of dextran have encouraged the proliferation of HFB-4 cells.

#### 3.2.2. Cellulose Composite with Synthetic Polymers

Bacterial cellulose is suitable for the fabrication of biopolymers, especially for healthcare-related applications, manufacturing scaffolds, implants, artificial blood vessels, and wound dressing materials, for wound and burn cases. The composite of bacterial cellulose with poly(vinyl alcohol) and hexagonal boron nitride was used for the preparation of a 3D scaffold for bone tissue engineering, using 3D printing technology. In composite bacterial cellulose, the major characteristics of the scaffold were determined. The addition of cellulose reduced the pore size in the composite and increased the viscosity to 81.3 mPa.s. Composites with bacterial cellulose have a lower tensile strength and the highest break in elongation, i.e., 93% was observed in the case of the composite prepared with 0.5% cellulose, 12% poly(vinyl alcohol), and 0.25% boron nitride. The biocompatibility assessment with human osteoblast cells suggested that at a lower concentration of cellulose, the viability of cells reduced by it further increased when the concentration of cellulose increased from 0.1/0.2% to 0.25 and 0.5%. The composite also offered a surface for cell adhesion and proliferation [92]. 

In a similar line, Zhang et al. [93] have fabricated macroporous hydrogels with dextran and polydopamine to be used as a carrier for antibiotics. Polydopamine affected the structure, as well as the functionality of the hydrogel, as with the increase in the concentration of dopamine, the pore size decreased and the surface area increased. The hydrogel has shown an increase in negative charge with the increase in the concentration of polydopamine in the composite, but the storage modulus and mechanical strength was increased. With the increase in dopamine content, the swelling ratio of hydrogel reduced and induced deswelling, which was due to the reduced pore size and higher interaction of dopamine with the internal structure of the hydrogel. The hydrogel lowered the viability of NIH3T3 cells slightly on the first day, but then no further cytotoxicity effect was reported. Hydrogels with a higher concentration of polydopamine were found to be suitable for drug loading and release. Hydrogels have shown up to 71% chlorhexidine acetate loading in 4.5 h, followed by its release of 12.58%, 16.06%, and 22.03% after 12, 24, and 48 h, respectively. The trend suggested that drug release was reduced with an increase in dopamine concentration. The work of Wang et al. [94] also emphasized the application of a cellulose composite for the preparation of artificial blood vessels with small diameters for thrombosis patients. A composite of poly(-caprolactone) and cellulose acetate was used to prepare nanofiber membranes for its further application in tubular scaffold fabrication, using different types of stainless-steel collectors. The composite scaffolds have water contact angles of 126.5° and 105.5°, which was increased by constructing a square-groove. In comparison to other collector mesh, scaffolds with a large mesh have 30% and 148% higher tensile strength over random-flat and tubular scaffolds, respectively. Similarly, the long mesh also offered 103% higher suture retention strength. Biocompatibility assessment revealed that the long mesh scaffold has 88% cell viability, and the blood coagulation index (BCI) was 5 min, which was around 89% of the standard value.

#### 3.2.3. Xanthan Composites with Synthetic Polymers

Extracellular polysaccharides are one of the most suitable materials for the preparation of scaffolds, grafts, and dressing materials, as they have water retention capacity, significant mechanical and tensile strength, and adaptability similar to the natural organs and tissues. In addition, it also supports cell adhesion, growth, and proliferation.

Cell growth has a direct influence on the conductivity and impedance of the medium, which may be used as an indirect method to assess and monitor cell growth within polymeric scaffolds. A conducting polymer composite was designed for the preparation of porous scaffolds with poly(3,4-ethylenedioxythiophene) and xanthan gum and compared with PEDOT: polystyrene sulfonate scaffolds (Figure 5). The semisynthetic composite scaffold carries important characteristics, such as the conductivity of poly(3,4-ethylenedioxythiophene), along with the biocompatibility and mechanical strength of xanthan gum. Composite scaffolds have interconnected pores, with a size range of 10–150 μm, that can be tuned as required, and Young’s module range was 10–45 kPa. The composite scaffold supports the cell growth of MDCK II eGFP and MDCK II LifeAct epithelial cells [95]. 

A series of composites were prepared with xanthan gum, citric acid, gelatin, glutaraldehyde, and HPLC-grade water, and evaluated for wound healing potential in rat skin wound models. All the composites have >90% of water-holding capacity. The presence of free and bound water was confirmed with FTIR studies, with inter- as well as intramolecular hydrogen bonds. Similar to water holding capacity, all the hydrogels have also shown significant wound healing capacity in deep second-degree skin burns in rats. After 20 days of application, a composite of xanthan gum–gelatin–glutaraldehyde and xanthan–citric acid and glutaraldehyde had a higher wound healing rate over others, as well as the control [96].

Malik and et al. [97] prepared an oral, drug carrier composite with chitosan, xanthan gum, monomer 2-acrylamido-2-methylpropane sulfonic acid, and potassium persulfate by free radicle polymerization technique. The components were crosslinked with N′ N′-methylene bis-acrylamide. The composite was loaded with acyclovir as a model antiviral drug commonly prescribed in herpes simplex virus infections. The composite had a higher thermal stability and a porous structure that encapsulated the drug compound. The composite hydrogel has shown pH-dependent drug releasing behavior. Ilomuanya et al. [98] prepared a hybrid composite with antibacterial, antioxidative, and anti-inflammatory behavior for wound dressing. The composite was prepared with poly(vinyl alcohol) and xanthan gum/hypromellose/sodium carboxymethyl cellulose. Composites with silver nanoparticles have higher antimicrobial potential and have inhibition of >99% against wound pathogens such as *Acinetobacter baumannii*, *E. coli*, *K. pneumoniae*, *P. aeruginosa*, *S. aureus*, *S. epidermidis*, and *Candida albicans*. Maximum bactericidal effect was observed from hypromellose nanocomposite, with 99.9% growth reduction, within 1 h of application. Dressing material prepared with composites also has free radical scavenging behavior, along with a reduction in the inflammatory response in RAW 264.7 macrophages. Animal model studies have confirmed that the hypromellose-based nanocomposite has a higher wound-healing process.

#### 3.2.4. Gellan Gum/Levan Composite with Synthetic Polymers

Composites with synthetic and natural components may have a higher applicability rate as natural polymers are biocompatible and biodegradable, but the addition of synthetic material may prolong the lifespan and modify the surface texture for the attachment of polymer. A levan-based composite was prepared with two different forms of levan, i.e., hydrolyzed and sulphated. Both hydrolyzed and sulfated levans were synthesized by microwave-assisted acid hydrolysis, with 5% acetic acid at 60% operating power for 60 s and mixing with chlorosulfonic acid for 24 h, respectively. The composite blend of 10% polycaprolactone (THF:DMF) and the aqueous solution of sulfated levan and hydrolysed levan in polyethylene oxide was used for coaxial electrospinning. The composite fiber has higher ultimate tensile strength and it increased with ShHL concentration. The composite increased the viability of L929 fibroblasts and HUVECs [99]. Adrover et al. [100] have prepared the composite beads for drug delivery with gellan gum, calcium ions (CaCl_2_), and with/without synthetic clay laponite ionotropic gelation technique. The addition of laponite reduced the swelling degree of polymeric beads. Gellan gum and gellan-silicate composite beads were loaded with theophylline and cyanocobalamin for in vitro release behavior. Laponite controlled the drug entrapment efficiency, as well as the slower drug release into the gastric environment. Theophylline followed Fickian behavior for drug release, while cyanocobalamin release behavior was greatly influenced by the physical/chemical interaction of the composite and drug.

For bone regeneration, an ideal membrane is of utmost importance that not only induces cell proliferation followed by mineralization, but is also able to withstand stress. A composite of calcium–poly-γ-glutamic acid and glycerol with gellan gum was studied for bone regeneration. In composite membranes, calcium aggregates are distributed uniformly and act as commendable performers in protein adsorption, and bone cell proliferation with MG63 cells. The hydrogel has promising results for bone repair, and γ-PGA acts critically. The exopolysaccharide member acts as biocompatible material, and γ-PGA improved cell adhesion and proliferation. It also increased the secretion of alkaline phosphatase and induced mineralization. The third member, ‘glycerol’, enhanced mechanical strength, elongation at break, as well as diffusion rate. Both glycerol and γ-PGA delayed the degradation of the composite [101]. The main contribution of synthetic polymers in the composite is to improve the tensile and mechanical strength and delay the degradation, an important feature that is required to provide sufficient time for tissue repair and recovery. 

#### 3.2.5. Pullulan Composites with Synthetic Polymers

Drug-delivery-related applications required hydrophobic carriers and functional groups that can offer a site for attachments. Exopolysaccharides such as pullulan are soluble in an aqueous environment. However, these are sensitive to modification by hydrophobic functional groups, including the cholesterol functional group, which make them amphiphilic and ensure the drug release.

Cholesteryl-modified aminated pullulan polymers were prepared with cholesterol succinate and pullulan. Succinic anhydride cholesterol (0.2–0.6 g), 0.18 g dimethylaminoaniline, 0.35 g 1-ethyl-(3-dimethylaminopropyl) carbodiimide salt, and acid salt were dissolved in DMSO at room temperature, which activated succinic anhydride cholesterol. The activated solution was added to 5.6% amino pullulan solution in DMSO and mixed at 50 °C for 48 h, followed by cooling to room temperature. Anhydrous ethanol was added to the reaction liquid that precipitated the composite. The composition of different components affects the properties of the composite. With respect to different concentrations of cholesteryl substitution, particle size reduced from 178.0, 144.4, and 97.8 nm, with an increase in the extent of substitution. With an increase in substitution, the hydrophobicity of the pullulan derivative increased and the particle size reduced. Hydrophobicity also influenced the drug release, as derivatives with maximum hydrophobicity have the slowest drug release, i.e., 57.8%; and the lowest hydrophobicity have maximum drug release, i.e., 72.7% after 48 h. In contrast, the efficacy against lung cancer cells increased with a reduction in hydrophobicity [102]. Mommer et al. [103] have functionalized the pullulan by thiolactone-based activation that adds amines. Modified pullulan offered the possibility of forming a composite with amine-containing biological substrates. With respect to thiolactone substitution (2.5/5.0 mol%), hydrogels have different mesh sizes, i.e., 27.8 and 49.1 nm respectively. Cell proliferation studies were conducted with different cell lines (normal human dermal fibroblasts and hepatocytes) and it was reported that gelatin and H-Gly-Arg-Gly-Asp-Ser-OH (GRGDS) support cell proliferation, while H-Gly-Arg-Gly-Asp-Ser-OH as well as H-Gly-His-Lys-OH acetate salt improved the proliferation of hepatocytes (HepG2) by up to 10 folds over gelatin. A thermosensitive composite was prepared with pullulan, carrying pendant carboxymethyl groups and amphiphilic triblock copolymer, poloxamer 407, by grafting. The final composite has about 83.8% poloxamer grafted on the copolymer. The composite was highly flexible and elastic, which allowed for the copolymer to regain and recover the native structure after the removal of external force/stimuli. The gel sustained amoxicillin release over a period of 168 h. The composite can be used as a hydrogel for scaffold fabrication, as well as for drug delivery [104].

High water holding and moisture retention with mechanical properties are the main strengths of bacterial nanocellulose suitable for the preparation of wound dressing and fabrication of biomedical device production, but cellulose lacks antimicrobial and wound-healing capacity. On the other hand, pullulan can contribute to wound healing, and zinc oxide nanoparticles offered antibacterial properties. Aminoalkylsilane grafted bacterial cellulose membrane and established an interconnection between cellulose with pullulan–ZnO nanoparticle hybrid electrospun nanofibers. Dressing materials prepared from composites have better blood clotting performance over the control, i.e., BNC. The composite released ZnO and offered 5 logs of higher antibacterial activity than cellulose. Cytotoxicity analysis with L929 fibroblast cells suggested that the composite was safe for fibroblast cell proliferation. In the animal (rat) model, the composite material offered rapid healing and re-epithelialization rate with the formation of a small blood vessel and synthesis of collagen [105].

**Table 2 polymers-15-01801-t002:** Exopolysaccharides composites in biomedical applications.

EPS Composites and Derivatives	Product	Applications	Preparation	Composite Properties	References
Gelatin- penta methyl cyclo pentadienyl triphenylphosphine ruthenium chloride, and sodium persulfate	Hydrogel	Wound healing	Gelatin solution was prepared in Milli-Q ultrapure water at 37 °C. The visible light photoinitiator, pentamethylcyclopentadienyl triphenylphosphine ruthenium chloride, and sodium persulfate (10 mM; Advanced BioMatrix, Inc., Sea Lion Pl, Carlsbad, CA, USA) were gently mixed with gelatin solution immediately before use	The minimum gelling time for hydrogel was 3.79 s for hydrogel with 10% hydrogel and 1 mM photoinitiator with LED intensity of 30 mW/cm^2^ Burst pressure 126.20 mmHg No toxicity against L929 fibroblast cells Wound closure of 72.9% in 7 days without inflammation	[106]
Poly(vinyl alcohol)/Dextran-aldehyde	Hydrogel	Wound dressing	PVA powder, mixed and swelled in water followed by homogenization at 90 °C for 2 h PVA and dextran aldehyde solution frozen at −20 °C for 6 h Lyophilize the sample to prepare PVA/DA hydrogel	Fluid adsorption capacity increased with dextran content Tensile strength decreased with the increase in dextran content Wound coverage was higher/equal to commercial products i.e., at 45% on day 4 and 68% on day 8 with hydrogel and 40% and 65% with commercial products	[107]
Gelatin-pullulan Composite Nanofibers	Nanofibers	Tissue engineering	Pullulan and gelatin solution were prepared in deionized water at 30 °C for 24 h The mixture was electrospinned at room temperature at DC voltage of 20 kV and pumping rate of 0.5 mL/h	Gelatin-pullulan composite prepared by mixing solutions on a magnetic stirrer at 30 °C for at least 24 h The surface tension of all compositions was below 70 mm/m Viscosity, conductivity, and surface tension increase with gelatin content but fiber diameter decreased Gelatin content also lowered the elongation at the break	[108]
P3HB4HB/(GE + PVA)	Scaffold	Tissue engineering	P3HB4HB solution in dichloromethane, used for electrospinning of core Gelatin and PVA solution in deionized water, used in shell	Pore size from 45–68 μm, porosity was 50–81% Water contact angle 68–83° Degradation increased with gelatine content Lower gelatine content higher will be the cell adhesion	[109]
Polycaprolactone/gelatin	Scaffold	Diaphragmatic muscle reconstruction	PCL mesh was designed with reinforcement of gelatin type A (porcine skin) and was prepared by 3D printing at 220 °C (400 μm diameter) at the rate of 8 mm/s	Fibers were hydrophilic with a pore size of 1–100 µm Mechanical strength 2.7–4 MPa Highest degradation with fibers having the highest gelatine content Fibre-supported adhesion and proliferation of cells	[110]
Gellan gum-egg shell membrane	Hydrogel	Regeneration of retinal pigment epithelium	Eggshells (washed with 0.05 M sodium carbonate) Cleaned samples were cryo-ground to powder (<100 µm) Powder added to 2% gellan gum solution and 0.03 *w*/*v*% calcium chloride at 70 °C	The compressive moduli of all combinations were 244 and 399 kPa The addition of 4% egg shale membrane reduced the cross-linking and viscosity by 40% Composite swelling also reduced by 30% Composite has no cytotoxicity against retinal pigment epithelium	[111]
Hydroxyapatite-embedded levan	Hydrogel	Dermal filler improved collagen production and anti-wrinkle activity	5% levan, 17% Pluronic F127, and 0.5 wt% of carboxymethyl cellulose was dissolved in deionized water Mixing at 4 °C for 24 h Addition of hydroxyapatite to the solution	0.1–5% hydroxyapatite (HAp)- levan composite hydrogel to improve in vivo collagen production and anti-wrinkle efficacy Increasing in HAp concentration reduced the in vitro stability and elasticity Composite has improved cell proliferation for human dermal fibroblast cells in comparison to levan alone The composite was stable over 8 week’s period. Composite with 1% HAp has excellent anti-wrinkle efficacy during 8 weeks by improving the collagen production	[112]
Alginate-gelatin	Hydrogel	Biomedical applications in wound dressing	4% aqueous solution of sodium alginate/gelatin at room temp and mixed for 4 h Addition of 5% oxidized starch to the solution followed by degassing under vacuum	Composite has maximum tenacity of 1.29 cn/dtex Maximum elongation of 4.41% (composite with 16.7% gelatin) Water absorption and retention increased by 19% (overall 335% and 311% respectively (composite with 16.7% gelatin)	[113]
Hydroxyapatite-chitosan-based hydrogels biomaterials loaded with metronidazole.	Hydrogel	Controlled drug delivery	1% Chitosan solution in acetic acid solution and 2.5% Metronidazole dispersion in deionized water was mixed for 1 h Na_2_HPO_4_, CaCl_2_ solution was mixed sequentially and pH adjusted to 10 with NH_4_OH After 1 h pH was adjusted to 7 and the addition of poly vinyl alcohol After 1 h mixing at 70 °C, the mixture was frozen at −18 °C overnight	Hydroxyapatite reduced the swelling ratio and prevent burst release with respect to pH variation Drug release was 38–73% as compared to the control High encapsulation efficiency (96%)	[114]
chitosan-gelatin scaffold loaded with aceclofenac	Scaffold	Controlled drug delivery	1% chitosan solution mixed with 3% (*v*/*v*) Tween 80 4% gelatin solution in distilled water was added to the chitosan solution at 37 °C. 0.2% *v*/*v* glutaraldehyde added as cross-linker 15 mg of Aceclofenac or [(2-{2,6-dichlorophenyl) amino} phenylacetooxyacetic acid] solution in methanol added dropwise to the CS-Tween 80	Porosity-100% allows scaffold degradation and sustainable drug release Best realised at pH: 6.8 (slightly acidic pH is good for drug release in osteopathic conditions) Become liquid in external pressure and reform gel when pressure removed No floccule in the scaffold and no aggregation behaviors	[115]
Curdlan-phosphorylated curdlan-ionic hydrogel-Metronidazole	Hydrogel	Controlled drug release	8% solution of curdlan and phosphorylated curdlan in NaOH Addition of 1,4-butanediol diglycidyl ether drop wise Sonication followed by lyophilisation	Swelling ratio increased: 9 g/g to 16 g/g Drug release increased from 50 to 90%	[116]
Hyaluronic acid-gelatin (0.5% HA-Ph + 5% gelatin-Ph)	Hydrogel	Adipose stem cells cultivation	Gelatin-Ph and hyaluronic acid-Ph solution prepared in phosphate-buffered saline The contents of [Ru(bpy)3]2+ and SPS in the solutions were fixed at 1.0 and 2.0 mM. hASCs were obtained from Lonza (Walkersville, MD, USA) and cultured in a growth medium for ASCs (PT-4505, Lonza) containing 10% (*v*/*v*) fetal bovine serum.	Diffusion coefficient: 1.2–0.8 (10^−10^ m^2^/s) Young modulus: 1.0–1.1 kPa Viscosity 80–90 mPa s	[117]
Dextran-Thyme Magnesium-Doped Hydroxyapatite	Coating	Antimicrobial coating	Ca(NO_3_)_2_·4H_2_O and Mg(NO_3_)_2_·6H_2_O solution was added into (NH_4_)_2_·HPO_4_ by drop-by-drop Precipitate was dispersed in 10% dextran and 1% thyme essential oil solution	As coating material Prevent biofilm formation by gram-positive and gram-negative bacteria and even *Candida*	[67]
Methacrylated gelatin-hyaluronic acid	Hydrogel scaffold	Tissue engineering	Methylated gelatin and hyaluronic acid solutions were prepared in PBS at 37 °C 0.5% (*w*/*v*) of Irgacure^®^ 2959 photoinitiator was added to solution	Hybrid hydrogel formed with modified gelatin and hyaluronic acid has a higher elastic modulus of 30 kPa, Porosity was around 91% in comparison to the control	[118]
‘Gelatin-hydroxy-phenyl propionic acid’- ‘hyaluronic acid tyramine’	Polymer network	Retinal ganglion cells replacement therapy	Gtn-HPA and HA-Tyr composites were prepared via carbodiimide/active ester-mediated coupling reaction in phosphate buffer saline H_2_O_2_ was added as a cross-linker	Gel point 151–167 s Elastic modulus 3101–4316 Pa Shear 835–1072 Pa	[119]
Gellan Gum, Alginate and Nisin-Enriched Lipid Nanoparticles	Hydrogel	Wound recovery	NSN-stearic acid nanoparticles fabricated by double emulsification/solvent evaporation method Hydrogel was prepared by mixing gellan gum, alginate, and nisin-stearic acid nanoparticle	Nisin-stearic acid-gellan gum-alginate nanoparticles were spherical Average particle size 300 nm Load size of 500 µg/mL was cytocompatible with L929 fibroblasts	[120]
Mg^2+^-Gellan Gum/Poly-Acrylamide	Hydrogel	Wound healing	1.5 wt% GG + 1.5 mol/L acrylamide monomer + 0.015 mol/L N,N′-methylene dia crylamide + 5 mmol/L ammonium persulfate mixing at 80 °C Cooled gel immersed in Mg^+2^ solution	Mg^2+^-GG/PAM hydrogel is 7.2 mol/m^3^ Tension strength 392 kPa The controlled release rate of Mg^2+^ (in PBS and alkaline conditions) Fibroblast cells have better proliferation over control Complete wound closure without scare	[121]
Oxidized gellan gum + carboxy methyl chitosan	Hydrogel	Drug delivery and wound dressing	Mixing of oxidized chitosan and gellan gum at room temp for 30 min Addition of CaCl_2_ at 37 °C	Increase in concentration of gellan gum microsphere increased decreased the gelation time Swelling rates were 35–34% (vary with components) G’ of Gel made with cellulose: gellan in ratio of 1:1, 2:1, and 3:1 was 1750 Pa, 2260 Pa and 2110 Pa, respectively. Drug release of 29.7% in 14 days	[122]
Gellan gum-alginate-calcium chloride	Hydrogel	Osteochondral repair	1% gellan gum solution in deionized water + 1.5% C alginate solution in deionized water + 1 mL CaCl_2_ mixing for 20–30 min	In osteochondral defect model with a diameter of 4.0 mm and depth 8.0 mm hydrogel-induced neovascularization in 4 weeks Repair the defect at week 8	[123]

The composite preparation and applications in different fields are still not very well developed. Some of them are under screening and others have crossed trials in cell lines. The efforts have received wide attention, and for outstanding findings, patents have been awarded (Table 3). 

Microbe-based polysaccharides are biocompatible, nonimmunogenic, biodegradable and most of these are USA FDA-approved and regarded as safe for human consumption [135]. According to GRAS notice 000099, the use of pullulan is allowed as an ingredient in tablets and capsules for dietary supplements [136]. The commercialization of such products still needs a long way to cover especially in the context of their behavior in various conditions to support their candidature. 

## 4. Challenges and Future Prospective

Microbial EPS and associated composites have offered several advantages such as biocompatibility, biodegradability, and nontoxicity; however, there is still no product available at the commercial level. The commercialization of products prepared from EPS composites needs to address some of the common challenges, summarized below.

### 4.1. Strain Selection 

The major fraction of EPS composites is microbial exopolysaccharides, which are produced by fermentation at various scales. Strain selection is one of the major problems associated with the large-scale production and commercialization of EPS and associated products. Several microorganisms have been screened for the production of EPS and some of them are either pathogenic or have low yield [137]. However, some of the microorganisms have offered compatible productivity without any pathogenicity, such as probiotic bacteria, including *Lactobacillus*, *Bifidobacterium*, and *Lactococcus*. In addition to yield, the tolerance to high salt concentrations (bile salt) and extremely acidic pH are other advantages that aided in the commercialization of products. Recent research has also proven that lactic acid bacteria also improved immunity and wound healing [49,138].

Genetic engineering of microorganisms is another possible strategy to improve productivity. However, the production of polysaccharides needs a combinatorial synthesis approach, as it is governed by multiple pathways. Cumulatively the pathways have the following three stages: nucleotide sugar precursors synthesis, extension, and synthesis of oligosaccharide units by glycosyltransferases and assembly of structural units to the final EPS, followed by export [139]. Some of the strategies have been summarized by Schmid et al. [140], but no report is available on the large-scale production of EPS with genetically engineered microorganisms. This may be due to the complex biosynthetic pathway governed by multiple gene systems. There is a need to work on this issue to increase EPS productivity with unique properties.

Production of exopolysaccharides, yield, and cost: Any microorganism can offer a higher possible production under optimal growth and production conditions. Before moving to higher-scale production, the identification of exopolysaccharides, their nature, and optimal conditions for maximum productivity are required [141]. As mentioned above, EPS production relies upon commercial-grade substrates with a higher cost, which also affects the product cost. A possible way out for the substrate is the utilization of low-cost lignocellulosic biomass from agricultural, as well as industrial sources, such as raw material. Previous research has used various agricultural biomass for pullulan production by *Aureobasidium pullulans*, i.e., hazelnut husk [142], sesame seed oil cake [143], corn cobs, and straw [144]. The process can be optimized via one factor or a statistical optimization approach. However, lignocellulosic materials, such as raw materials, need preprocessing such as hydrolysis, followed by detoxification. Acid hydrolysis is one of the most common methods used for sugar recovery in hydrolysate, but some byproducts, including furfurals and hydroxymethyl furfurals (from pentose and hexose), were also produced along with phenolics from lignin. These byproducts have shown an inhibitory effect on microbial growth and hence hydrolysate was detoxified by activated charcoal and membrane separation [33,145,146]. These additional steps for biomass processing and hydrolysate preparation increased the process, as well as product cost. The use of tolerant microorganisms might help overcome the challenges associated with inhibitors.

### 4.2. Downstream Processing

Downstream processing and product recovery is one of the major challenges, as it contributes to the majority of the product cost. Therefore, an efficient and low-cost downstream process is a prerequisite for the commercialization of exopolysaccharides and associated products [141]. The chemical nature and characteristics of exopolysaccharides, such as high viscosity and gel-forming ability, hinder their extraction. Some of the recovery approaches are trichloroacetic acid and proteases-assisted protein removal, dialysis, chromatography, and solvent-based precipitation used for EPS recovery. The ultra-grade of purity of a product is necessary for biomedical applications, hence the selection of the recovery process should be done as per the target product, as well as the physicochemical properties of EPS. Proteins are removed by protease/trichloroacetic acid to prevent contamination and cross-reaction. However, heat or use of concentrated acid damage the native EPS structure and change the integrity of bonds, branching, and sugar monomers. [147].

### 4.3. Composite-Forming Ability of EPS

Usually, EPSs are water-soluble (except a few, such as cellulose) and hydrophilic, which makes EPS prone to degradation in storage, as well as in the cellular environment. This also influences the selection of drug and secondary polymers, as components with similar charges usually make an unstable system due to repulsion. Due to the presence of hydroxyl and carboxylic functional groups, most of the EPS molecules are negatively charged, and thus loading of negatively charged groups and blending with negatively charged polymers is not possible due to electrostatic repulsion [137,148,149]. 

Derivatization of exopolysaccharides is one of the most common approaches to change the net charge, as well as improve interaction with other molecules, as observed in the case of cellulose, which is insoluble in water as well as organic solvents. The availability of the hydroxyl functional group makes it possible to modulate its chemical nature, including solubility, hydrophilicity/hydrophobicity, and mechanical strength either, by degradation or derivatization. Cellulose ether is the derivative prepared by replacing the hydroxyl group with the hydrocarbon group. It includes carboxymethyl, methyl, hydroxypropyl methyl, and hydroxyethyl derivatives of cellulose. Derivatization of cellulose has improved thermo-plasticity, apart from hydrophilicity of derivative, in comparison to native cellulose [150]. Similar changes were also observed in gellan gum hydrogel used for muscular injury. The addition of laminin protein improved the muscular tissue auto-healing capacity, but 3D hydrogel needs extensive crosslinking and porous structure. To attain a stable porous interconnected structure, gellan gum was derivatized with divinyl sulfone. The composite of gellan gum–divinyl sulfone derivative and gellan gum was used to prepare 3D hydrogel and functionalized with laminin-derived peptide. The composite was encapsulated with skeletal muscle cells. The modification of gellan gum with divinyl sulfone improved cross-linking that stabilized the 3D framework of hydrogel [151].

### 4.4. Stability and Degradation Products of EPS Composites

Exopolysaccharide composites are comprised of EPS and natural or synthetic polymers. EPS mainly has polysaccharides as a major component, while natural polymers may also have proteins and lipids, etc. Each biopolymer, including sugar/carbohydrate, proteins, lipids, and nucleic acids, is prone to degradation and may endure only a few hours to a few weeks (depending upon the environment or type of polymer). The study conducted by McClatchy et al. [152] also proved that the stability of proteins depends upon the cellular environment. One of the advantages of the use of composite scaffolds made with natural polymers is the complete degradation of EPSs such as pullulan, dextran, etc., as well as secondary polymers, into CO_2_ and H_2_O. Some nitrogen-containing polymers, such as chitin and chitosan, release amino sugars which can be utilized by microorganisms and living cells [153,154]. To overcome the challenges associated with stability, synthetic polymers can be used instead of natural polymers. Some of the common synthetic polymers used in biomedical applications are polyarylsulfones, polysulfone, polyvinylpyrrolidone, polyamide, polycarbonate, polyacrylonitrile, PMMA, polyester polymer alloy, ethylene vinyl alcohol copolymers; and molecular-thin nanoporous silicon membranes are the common synthetic polymers used in hemodialysis membranes; polymethyl pentene in extracorporeal membrane oxygenation; polyester, polyether, and polycarbonate-based polyurethanes in catheters; nylon, polyurethanes with acrylate in wound dressing; fibrin glue in sealants and adhesive; expanded polypropylene, polytetrafluoroethylene, polyethylene terephthalate, polyvinylidene difluoride membrane in surgical meshes; and poly-tetrafluoroethylene, polyethylene terephthalate, dacron, nylon in scaffold and ligament repair [155]. It has been proven that the addition of synthetic polymer has improved the mechanical and woven strength of natural exopolysaccharides. Kenawy et al. [91] have shown that the addition of poly(vinyl alcohol) (PVA)to dextran has improved the mechanical stability of the composite for electrospinning. Further addition of citric acid as a cross-link had a conducive effect on the mechanical, as well as tensile strength and stability of the composite due to extensive cross-linking. 

### 4.5. Side Effects of Synthetic Polymers

As discussed in the above section, synthetic polymers have positive results when blended with exopolysaccharides, but these synthetic polymers have shown side effects in a living system. Much literature has emphasized side effects and immunogenic responses with synthetic polymers. Acrylates are used in baby diapers, and prostheses [156], but acrylates may lead to serious allergic responses and dermatitis [157]. Similar kinds of side effects have also been observed with other synthetic polymers and adhesives, including anaphylaxis allergic reaction and arachnoid plasty in the case of fibrin glue [158,159], volitional swallowing, gastrointestinal obstruction, asthma, and allergy with polyethylene terephthalate [160] and oxidative stress from polypropylene used in mesh surgery [161]. On the other hand, some polymers have offered advantages over others, as their degradation product can be utilized in cells as in the case of poly-lactides. These kinds of polymers release lactic acids and respective oligomers upon degradation, which can be utilized by the cell during normal metabolism [162]. 

The major challenges associated with the commercialization of exopolysaccharide-based composites are its bulk-scale production, cost-effective recovery, and detailed analysis for optimum storage and transit conditions. However, there is no literature available that provides any information about the degradation of EPS composites, suitability of application, commercial cost of the products, and appropriate stability profile.

## 5. Conclusions

Exopolysaccharides have proven their candidature in the biomedical and healthcare sector mainly attributed to their biocompatibility, nontoxicity, and degradability. However, their limited mechanical and tensile strength, along with solubility in different solvents, obstructs their commercialization. Synthetic polymers have higher stability and strength, but are disadvantaged by side effects and compatibility issues. In order to achieve both compatibilities as well as tunability in strength, adaptability, and stability, composites are preferred over EPSs. The blending of natural and synthetic polymers might improve the physical and chemical characteristics. The composites have offered higher tensile and mechanical strength, along with water retention, slower degradation, drug-carrying capacity, and compatibility for biological applications, including 3D scaffold and wound dressing material fabrication, drug carrier properties, and biomedical sealant potential. Composites can support biological tissue and support healing due to improved adhesion and cell proliferation. The commercialization of composites needs in-depth study regarding stability in storage and transport, degradation behavior, and cost of the final product.

## Figures and Tables

**Figure 1 polymers-15-01801-f001:**
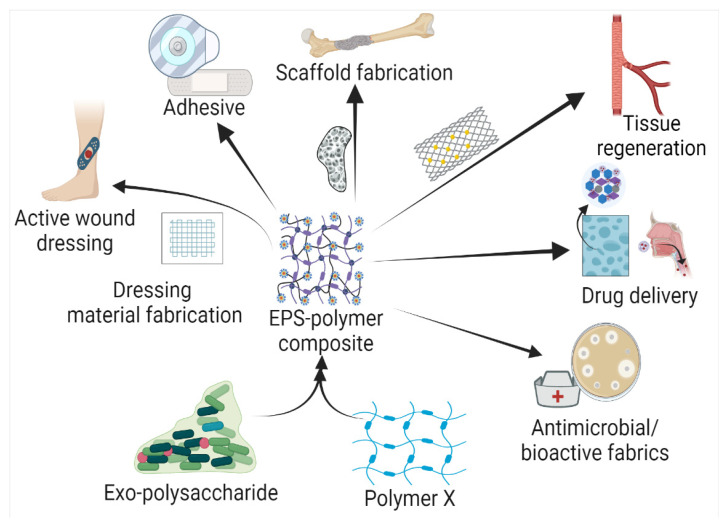
Exopolysaccharide composites with natural and synthetic polymers and their application in the health sector.

**Figure 2 polymers-15-01801-f002:**
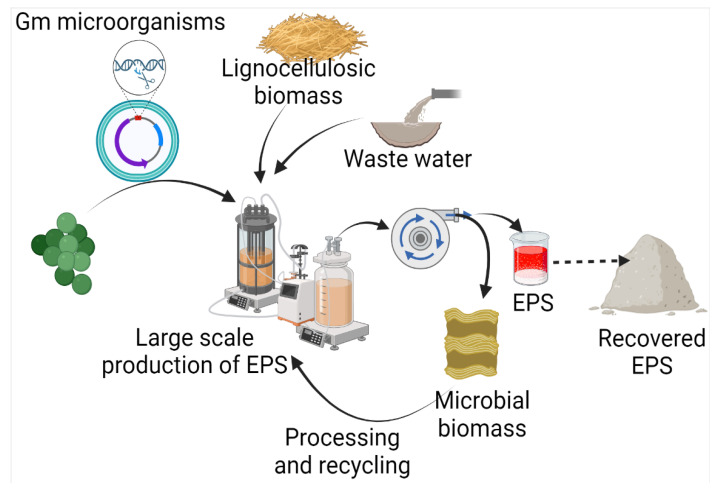
Microbial production of EPS using waste, and its recovery.

**Figure 3 polymers-15-01801-f003:**
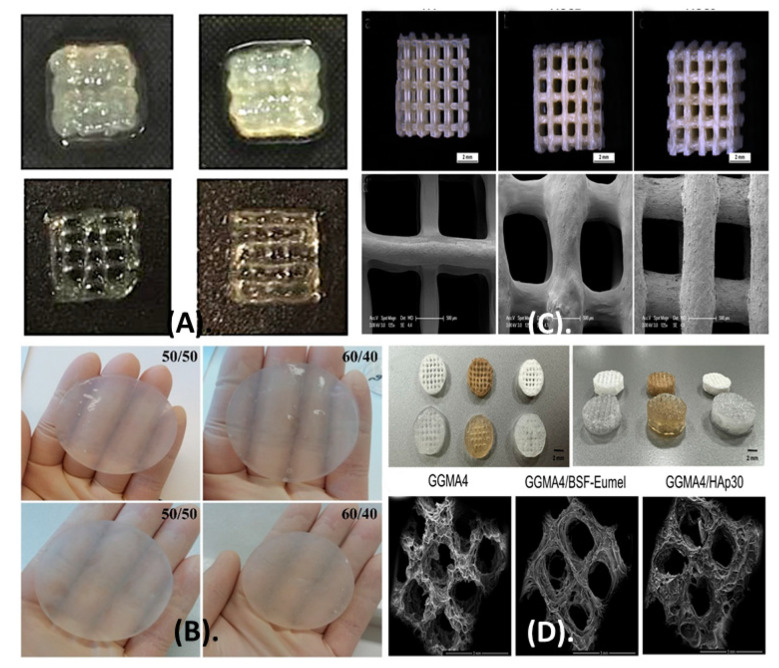
Composite scaffold prepared from different exopolysaccharides (**A**). Xanthan gum–gelatin [51]; (**B**). Xanthan gum–Konjac glucomannan [52]; (**C**). Alginate–gellan gum composites [53] and (**D**). Composite methacrylated gellan gum [54].

**Figure 4 polymers-15-01801-f004:**
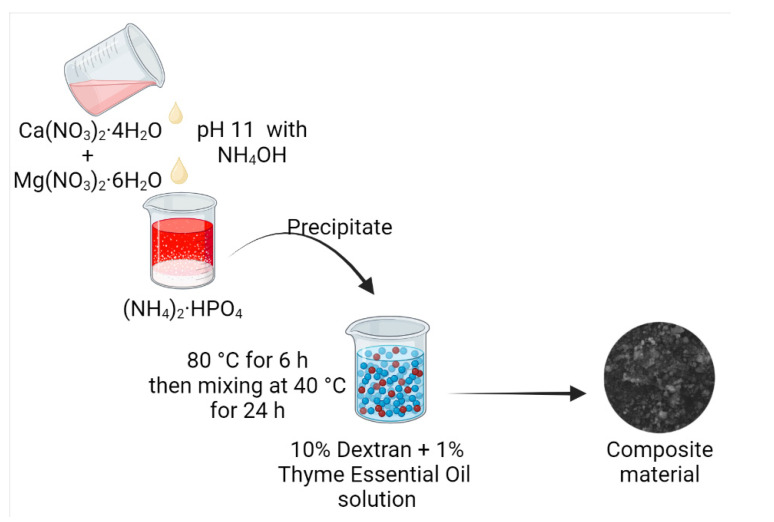
Preparation of antimicrobial coatings with dextran, thyme essential oil, magnesium salt, and hydroxyapatite, as suggested by Iconaru et al. [67].

**Figure 5 polymers-15-01801-f005:**
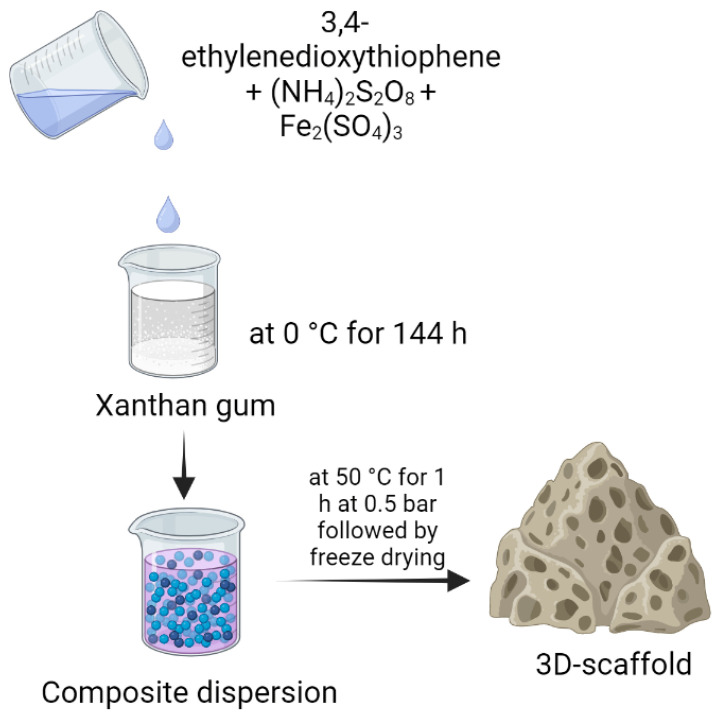
3D scaffold prepared with xanthan gum and poly(3,4-ethylenedioxythiophene), as described by del Agua et al. [95].

**Table 3 polymers-15-01801-t003:** Patents awarded in microbial exopolysaccharide-related research.

Patent ID	EPS	Microorganisms	Application	Reference
WO2023036938A1	EPS	*Firmicutes*	Gene mutation in *Firmicutes* for EPS production	[124]
WO2023038519A1	Biopolymers	*aerobic granular sludge and anam- mox granular sludge*	Modification of biopolymers using polyols and polyacids	[125]
WO2023025235	EPS	*Lactic acid bacterium*	Treatment of sleeping disorder	[126]
WO2023020880	EPS	*Paenibacillus* strains	Production	[127]
RU2782953	-	*Paenibacillus polymyxa* strain essutm-2	Production	[128]
WO2023275343	Low-molecular-weight he800 exopolysaccharide derivatives	*Vibrio diabolicus* HE800	Anti-cancer	[129]
RU2784088	-	*Paenibacillus polymyxa* vsgutu-1	Production	[130]
WO2023014213	Cellulase	*Acetobacter aceti*	Production	[131]
CN115572690A	EPS	Lactic acid bacteria	Selection of EPS producing bacterial	[132]
CN115554197A	EPS	-	Soothing and repairing based cosmetic formulations	[133]
CN115505610A	EPS	*Chlorella pyrenoidosa*	Antitumor activity	[134]

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
