# Peer review of "Microbial Exopolysaccharide Composites in Biomedicine and Healthcare: Trends and Advances"

_polymers, 2023, doi:10.3390/polym15071801_

Round 1

Reviewer 1 Report

The presented manuscript has reviewed microbial exopolysaccharides composites as healthcare products. This paper is well organized, and I recommend this paper for publication after a minor revision;

- It is suggested to discuss about the advantages and drawbacks of EPSs in a separate section.

- It is suggested to add some detailed information about the fabrication procedures of microbial exopolysaccharides-based products for biomedical applications.

- It is suggested to add a column in Table 2 to introduce the final form of EPS-based product. For example hydrogel, fiber, fabric …

- It is suggested to discuss about the commercial marketing of the EPS-based products at future perspective section.

- As the authors also mentioned, there are several published works focused on the production of EPS composites for biomedical applications. It is suggested to add some images from the produced EPS composites for biomedical application.

- There are some typo or grammar errors in the text, which is suggested to be doublechecked.

Author Response

Reviewer #1

The presented manuscript reviewed microbial exopolysaccharides composites as healthcare products. This paper is well organized, and I recommend this paper for publication after a minor revision;

Response: We appreciate the reviewer’s efforts and comments to improve the standard of our manuscript.

R1Q1: It is suggested to discuss about the advantages and drawbacks of EPSs in a separate section.

R1A1: Thank you for your suggestion. As we have organized our manuscript in various sections based on types of EPS and their composite, different types of EPS and composites have different advantages and disadvantages, so this information has been discussed in their respective section. The preparation of a new section focused on advantages and disadvantages will disrupt the flow so we have decided to keep it as it is. The advantages and disadvantages of different exopolysaccharides have been discussed in the introduction and respective applications of EPSs with synthetic and natural polymers. The sections have been highlighted.

In the introduction line no 85-90 (new information added)

The main bottleneck of using native EPS molecules in commercial products are solubility in different mediums, bioavailability, degradation, etc. xanthan, a HEEPS comprised of glucose backbone along with trisaccharide side chains and has poor thermal stability and electrical conductivity and is prone to microbial contamination [15,16]. Curdlan has high immunomodulatory potential, good gelling ability, and thermal stability but suffers from the issue of solubility in water [17]. Hyaluronic acid has high water retention but poor mechanical stability [18,19].

In addition, advantages and disadvantages have been discussed in respective applications.

Dextran: Section 3.1.2; Page 11; Line 306-312

Xanthan: Section 3.1.3; Page 12; Line 362-365

Pullulan: Section 3.1.4; Page 13; Line 435-441

Levan: Section 3.1.5; Page 14; Line 491-497

Gellan: Section 3.1.6; Page 15; Line 538-540

R1Q2: It is suggested to add some detailed information about the fabrication procedures of microbial exopolysaccharides-based products for biomedical applications.

R1A2: Authors are grateful for the suggestion. In Table 2 we have discussed various examples of EPS and its composite materials with their synthesis methods and applications. We have added more detailed information and discussed some more examples in various sections as follows.

Section 3.2.4 Line 694-701

Both hydrolyzed and sulfated levans were synthesized by microwave-assisted-acid hydrolysis with 5% acetic acid at 60% operating power for 60 sec and mixed with chlorosulfonic acid for 24 h respectively. A composite blend of 10% polycaprolactone (THF:DMF) and the aqueous solution of sulfated levan and hydrolysed levan in polyethylene oxide was used for coaxial electrospinning. Composite fiber has higher ultimate tensile strength and it increased with ShHL concentration. The composite increased the viability of L929 fibroblasts and HUVECs [99].

Section 3.2.5 Line 731-745

Cholesteryl-modified aminated pullulan polymers were prepared with cholesterol succinate and pullulan. 0.2-0.6 g succinic anhydride cholesterol, 0.18 g dimethylami-noaniline, 0.35 g 1-ethyl-(3-dimethylaminopropyl) carbodiimide salt, and acid salt was dissolved in DMSO at room temperature which activated succinic anhydride cholesterol. The activated solution was added to 5.6% amino pullulan solution in DMSO and mixed at 50 °C for 48 h followed by cooling to room temperature. Anhydrous ethanol was added to the reaction liquid that precipitated the composite. The composition of different components affects the properties of the composite. With respect to different concentrations of cholesteryl substitution, particle size reduced from 178.0, 144.4, and 97.8 nm with an increase in the extent of substitution. With an increase in substitution, the hydrophobicity of the pullulan derivative increased and particle size reduced. Hydrophobicity also influenced the drug release as derivatives with maximum hydrophobicity have the slowest drug release i.e., 57.8% and lowest hydrophobicity have maximum drug release i.e., 72.7% after 48 h. In contrast, the efficacy against lung cancer cells increased with a reduction in hydrophobicity [102].

R1Q3: It is suggested to add a column in Table 2 to introduce the final form of EPS-based product. For example hydrogel, fiber, fabric …

R1A3: A separate column has been added to specify the final product.

R1Q4: It is suggested to discuss about the commercial marketing of the EPS-based products at future perspective section.

R1A4: The most of studies and information we were able to collect from online resources showed that EPS-polymer composites production and applications study limited to lab-scale only and there is no product available at the commercial level. It needs more effort regarding the safety and stability studies for commercialization. The information related to this is summarized in ‘Section 4’.

R1Q5: As the authors also mentioned, there are several published works focused on the production of EPS composites for biomedical applications. It is suggested to add some images from the produced EPS composites for biomedical applications.

R1A5: We have added some images related to EPS composite biomedical applications Fig 3 and also discussed in the text. Piola et al., [51] prepared composite hydrogel with gelatin and xanthan gum to support the growth of human skin cells. The composite was printed with CellInk Inkredible 3D printer using glutaraldehyde solution as crosslinker. The printed hydrogel was compatible and suitable for the growth of human keratinocyte as well as fibroblast. Alvel et al., [52] also prepared a composite hydrogel of xanthan with Konjac Glucomannan which was also focused on wound healing. On the other hand 3D scaffold for tissue engineering was prepared with alginate-gellan gum [53] and methacrylated gellan gum [54].

R1Q5: There are some typo or grammar errors in the text, which is suggested to be doublechecked.

R1A5: We have tried our best to remove all the typos and grammatical errors using spell check and Grammarly.

Reviewer 2 Report

1. There is no outlook in the Abstract.

2. If the image is a published document, please note the copyright acquisition.

3. There are many repeated statements. Please condense the text,like Page 3 line 106-109 ï¼›Page 8 line 178-179.

4. Conclusion should be described in detail.

5. Suggesting author provides more figures about synthesis of composites, In addition, the chemical structure of some microbial exopolysaccharides should be showed.

Author Response

Reviewer #2

R2Q1: There is no outlook in the Abstract.

R2A1: We appreciate reviewers’ efforts and comments to improve the standard of our manuscript. The abstract has been updated and future possibilities have been added. Information added as “However, the commercialization of these products stills needs in-depth research considering the commercial aspects like stability under ex-vivo and invivo environments, in presence of biological fluids and enzymes, degradation profile and interaction within living systems. The opportunities and potential applications are diverse but need more elaborative research to address the challenges”.

R2Q2: If the image is a published document, please note the copyright acquisition.

R2A2: The images used in this manuscript designed by us and some newly added figures are adopted from other articles are under creative common licenses. We have mentioned the reference from where these figures has been adopted.

R2Q3: There are many repeated statements. Please condense the text like Page 3 line 106-109; Page 8 line 178-179.

R2A3: The manuscript has been checked and revised to remove the error and redundancy.

R2Q4: Conclusion should be described in detail.

R2A4: We appreciate the reviewer’s comments. The conclusion has been elaborated as suggested. Exopolysaccharides have proved their candidature in the biomedical and healthcare sector mainly attributed to their biocompatibility, nontoxicity, and degradability. However, their limited mechanical and tensile strength along with solubility in different solvents obstructed the commercialization. Synthetic polymers have higher stability and strength but suffer from side effects and compatibility issues. In order to achieve both compatibilities as well as tunability in strength, adaptability, and stability the composites are preferred over EPS. The blending of natural and synthetic polymers might improve the physical and chemical characteristics. The composites have offered higher tensile and mechanical strength along with water retention, slower degradation, drug carrying capacity, and compatibility for biological applications including 3D scaffold and wound dressing material fabrication, drug carrier, and biomedical sealant. The composites can support biological tissue and support healing due to improved adhesion and cell proliferation. The commercialization of composites needs in-depth study regarding stability in storage and transport, degradation behavior, and cost of the final product.

R2Q5: Suggesting author provides more figures about synthesis of composites,

R2A5: We have added more information and figure on this (fig 4 and 5).

R2Q6: In addition, the chemical structure of some microbial exopolysaccharides should be showed.

R2A6: The structures of microbial exopolysaccharides have been added in Table 1 for better understanding.

Reviewer 3 Report

In this review:

The advancements on the application of exopolysaccharides in the health therapy are reported.

The paper contains attractive information for a variety of readers.

I have somme comments that should be addressed.

1. Some paragraphs need to be revised in order to improve the reading like those from line 124 to 132;

2. In Table 2, the EPS and the second polymer (the load) should be specified.

3. Insert reference number after the author is cited; for example, in line 212, line 351, and so on.

4. In section 3.1.3, reported compressive strenght; how this parameters compare with another cases?

5. In line 343, correct the angle units.

6. It is recommende a Table speccifying the composite forming ability and the challenges in combination with downstream processing, even the degradation.

7. Conclusions me be extended.

6. 

Author Response

Reviewer #3

In this review: The advancements on the application of exopolysaccharides in the health therapy are reported. The paper contains attractive information for a variety of readers. I have some comments that should be addressed.

Response: We appreciate reviewer’s efforts and comments to improve the standard of our manuscript.

R3Q1: Some paragraphs need to be revised in order to improve the reading like those from line 124 to 132;

R3A1: The manuscript has been revised to remove the error and redundancy.

R3Q2: In Table 2, the EPS and the second polymer (the load) should be specified.

R3A2: We have added more information in table Table 2 to make it more informative.

R3Q3: Insert reference number after the author is cited; for example, in line 212, line 351, and so on.

R3A3: We have revised and formatted reference citations.

R3Q4: In section 3.1.3, reported compressive strength; how this parameter compares with another cases?

R3A4: We have added the suggested information. Composite with nanocrystals and nanotubes have porosity ranging from 91.7 ± 0.81% to 88.5 ± 0.64% and water uptake capacity ranging from 14.73.7 ± 0.46 g/g to 11.34 ± 0.32 g/g. The composite was thermally stable and had high compressive strengths of 91.1 ± 1.2 kPa to 114.4 ± 0.6 kPa in dry form and 9.0 ± 0.8 kPa to 10.6 ± 0.8 kPa in wet form. The composite has high cytocompatibility for MC3T3-E1 osteoblastic cells and the viability of cells increased with nanotube components in the composite. Both nitrocellulose and nanotubes increased the mechanical stability of the composite along with conducive bioactivity including higher cell adhesion and proliferation. The compressive strength of composites was higher than alginate alone as well as alginate-xanthan gum blend i.e., 91 kPa and 80.7 kPa. The blending of xanthan gum lowered the stiffness which was further improved by the addition of nanocrystals and nanotubes and maximum strength of 114.4 kPa [70].

R3Q5: In line 343, correct the angle units.

R3A5: The error has been rectified and the unit has been changed to superscript.

R3Q6: It is recommended a Table specifying the composite forming ability and the challenges in combination with downstream processing, even the degradation.

R3A6: The major challenges have been summarized in different sections and creating a table for the same data add redundancy in the manuscript. This information is already discussed various section as follows:

Section 4.2 Downstream processing

Section 4.3 Composite forming ability of EPS

Section 4.4 Stability and degradation products of EPS composites

Section 4.5 Side effects of synthetic polymers

Reviewer 4 Report

The authors have presented excellent collection on microbial exopolysaccharides as composite forming materials for biomedical application, suggested some addition/correction to improve the manuscript. 

1. Suggested to add microbial exopolysaccharides characterization techniques
2. Regulatory aspect of microbial exopolysaccharides
3. Microbial exopolysaccharides applications in other or novel drug delivery as miscellaneous section
4. Suggested to tabulate patent information on microbial exopolysaccharides
5. Depth review require addition of previous clinical application of composite formulated using microbial exopolysaccharides and utilised for biomedical applications. so suggested to add permitted copyright images to make more interesting this manuscript for readers.

Good Luck

Author Response

Reviewer #4

The authors have presented an excellent collection on microbial exopolysaccharides as composite forming materials for biomedical application, suggested some addition/correction to improve the manuscript.

Response: We appreciate reviewers’ efforts and comments to improve the standard of our manuscript.

R4Q1: Suggested to add microbial exopolysaccharides characterization techniques.

R4A1: The various techniques for EPSs characterization have been added in Section 2 (Page 8; Line 178-195). Preparation of composite for healthcare applications needs high purity therefore the downstream processing becomes an inseparable part of processing after fermentation. After production, recovery identification of structural and chemical characteristics are necessary for further applications. The characterization of EPSs is quantitative as well as qualitative. As EPSs are mainly comprised of carbohydrates, conjugated with other biomolecules, hence basic characterization techniques employed are colorimetric estimation and use of spectrophotometry [49]. For carbohydrate estimation ‘Dinitrosalicylic Acid Reagent’ is one of the common methods which quantify the reducing sugars [50]. Similarly, for protein Bradford’s dye-binding method [51] and Lowry's method are used [52]. Besides, basic characterization with colorimetric methods, Fourier Transform Infrared Spectroscopy (FTIR) is employed to detect the available functional groups and structural functionalities of EPSs. For the detailed structure of EPSs, techniques like Nuclear magnetic resonance (NMR) and Mass spectrum are used. In NMR sample is dissolved in deuterated solvents for quantification with respect to internal standards [53,54]. The mass spectrum of EPS provides monosaccharide composition. For analysis, EPSs are hydrolyzed with acid hydrolysis followed by silylation derivatisation. The derivatives are detected and identified by gas chromatography-mass spectrometry [53]. The biological potential of EPS has followed the general procedure for antimicrobial, antioxidant, anti-inflammatory and other activities [49].

R4Q2: Regulatory aspect of microbial exopolysaccharides

R4A2: Thank you for your suggestion. There is no separate regulatory related guidelines and literature available for EPS composite materials and their applications in biomedical and health sector, but a long history of research shows EPS is safe for use in human food and biomedical purpose. We able to find some information related to EPS and included in the manuscript as “Microbes based polysaccharides are biocompatible, nonimmunogenic and biodegradable and most of these are USA-FDA approved and regarded as safe for human consumption [136]. According to GRAS notice 000099 the use of pullulan is allowed as ingredient in tablets and capsule for dietary supplements [137]. The commercialization of such products still needs a long way to cover specially in the context of their behavior in various conditions to support their candidature.

R4Q3: Microbial exopolysaccharides applications in other or novel drug delivery as miscellaneous section

R4A3: The drugs are indeed one of the most important parts of healthcare. The applications of EPSs and composite have been summarized and categorized based on the EPS compounds with natural and synthetic polymers and several examples of EPS composite materials have been already discussed in the respective section. Creation of separate sections based on application will disrupt the architect of article. Some of the applications of EPSs have been added in introduction (Line 75-85). Besides, the majority of applications in healthcare are still under trial. Prasher et al., [11] used dextran derivative i.e., acetylated dextran as drug-delivery vehicle for the treatment of respiratory disease due to biodegradability, pH sensitivity, high encapsulation efficacy, and ability to cross the mucosal layer. Yahoum et al., [12] encapsulated metformin hydrochloride in xanthan gum microspheres and found that sustainable release of metformin hydrochloride from microsphere. The eyes are one of the most sensitive parts of the body that need special care. EPSs have proved safe and biocompatible for biological systems which allow their application in ophthalmic formulations. Khare et al., [13] evaluated an ophthalmic solution comprised of gellan gum-based nanosuspension with posaconazole for fungal keratitis.

R4Q4: Suggested to tabulate patent information on microbial exopolysaccharides

R4A4: The patents associated with EPSs production and applications have been summarized in table 3.

R4Q5: Depth review require addition of previous clinical application of composite formulated using microbial exopolysaccharides and utilized for biomedical applications. so, suggested to add permitted copyright images to make more interesting this manuscript for readers.

R4A5: The authors appreciate the suggestion. The images of previously prepared composite materials have been added in fig 3.

Reviewer 5 Report

The present review reports the applications of microbial exopolysaccharides (EPS) and their composites with natural and synthetic polymers in biomedicine and healthcare. In order to improve the quality of the review I recommend minor revision before publication.

My remarks are detailed below:

Please, check carefully your English – especially the use of singular and plural forms, as well nouns or adjectives; for example, in title use biomedicine instead of biomedical; composite or composites (3.1 Exopolysaccharide compositeS with natural materials, 3.2 Exopolysaccharide compositeS with synthetic materials), etc.

When you cite references use “et al.” instead of “colleagues”

The correct chemical name of PVA is poly(vinyl alcohol)

The Sections 3.2.1 and 3.2.2 describes the EPS composites with synthetic polymers. However, there is mentioned montmorillonite (natural clay mineral, not polymer) and graphene oxide (single-atomic-layered form of graphene, not polymer). Please, remove it or change the title of the sections.

Author Response

Reviewer #5

The present review reports the applications of microbial exopolysaccharides (EPS) and their composites with natural and synthetic polymers in biomedicine and healthcare. In order to improve the quality of the review I recommend minor revision before publication.

Response: We appreciate the reviewers’ efforts and comments to improve the standard of our manuscript.

R5Q1: Please, check carefully your English – especially the use of singular and plural forms, as well nouns or adjectives; for example, in title use biomedicine instead of biomedical; composite or composites (3.1 Exopolysaccharide composites with natural materials, 3.2 Exopolysaccharide composites with synthetic materials), etc.

R5A1: The authors are grateful for the suggestion. We have checked it carefully to avoid any grammatical mistakes.

R5Q2: When you cite references use “et al.” instead of “colleagues”

R5A2: We have made corrections throughout the manuscript.

R5Q3: The correct chemical name of PVA is poly(vinyl alcohol)

R5A3: The name has been updated across the manuscript.

R5Q4: The Sections 3.2.1 and 3.2.2 describes the EPS composites with synthetic polymers. However, there is mentioned montmorillonite (natural clay mineral, not polymer) and graphene oxide (single-atomic-layered form of graphene, not polymer). Please, remove it or change the title of the sections.

R5A4: Thank you for pointing out our mistake. We have removed this information from the manuscript.

Round 2

Reviewer 2 Report

the current manuscript can be accepted.

Reviewer 3 Report

The new versin of the paper has been improved adecuately; I think, it should be accepted.